# ZARTS: On Zero-order Optimization for Neural Architecture Search

**Xiaoxing Wang**[1], **Wenxuan Guo**[1], **Jianlin Su**[2], **Xiaokang Yang**[1], **Junchi Yan**[1*]
[1]Department of Computer Science and Engineering, Shanghai Jiao Tong University
[2]Shenzhen Zhuiyi Technology Co., Ltd.
`{figure1_wxx,arya_g,xkyang,yanjunchi}@sjtu.edu.cn`
`bojonesu@wezhuiyi.com`

## Abstract

Differentiable architecture search (DARTS) has been a popular one-shot paradigm for NAS due to its high efficiency. It introduces trainable architecture parameters to represent the importance of candidate operations and proposes first/second-order approximation to estimate their gradients, making it possible to solve NAS by gradient descent algorithm. However, our in-depth empirical results show that the approximation often distorts the loss landscape, leading to the biased objective to optimize and, in turn, inaccurate gradient estimation for architecture parameters. This work turns to zero-order optimization and proposes a novel NAS scheme, called ZARTS, to search without enforcing the above approximation. Specifically, three representative zero-order optimization methods are introduced: RS, MGS, and GLD, among which MGS performs best by balancing the accuracy and speed. Moreover, we explore the connections between RS/MGS and the gradient descent algorithm and show that our ZARTS can be seen as a robust gradient-free counterpart to DARTS. Extensive experiments on multiple datasets and search spaces show the remarkable performance of our method. In particular, results on 12 benchmarks verify the outstanding robustness of ZARTS, where the performance of DARTS collapses due to its known instability issue. Also, we search on the search space of DARTS to compare with peer methods, and our discovered architecture achieves 97.54% accuracy on CIFAR-10 and 75.7% top-1 accuracy on ImageNet. Finally, we combine our ZARTS with three orthogonal variants of DARTS for faster search speed and better performance. Source code will be made publicly available at: `https://github.com/vicFigure/ZARTS`.

## 1 Introduction

It remains open to search for efficient architectures automatically instead of by humans [27, 14, 15]. Neural architecture search (NAS) has attracted wide attention, which can be modeled as bi-level optimization for network architectures and operation weights. One-shot NAS [1] is a popular search framework that regards neural architectures as directed acyclic graphs (DAG) and constructs a supernet with all possible connections and operations in the search space. DARTS [20] further introduces trainable architecture parameters to represent the importance of candidate operations, which are alternately trained by SGD optimizer along with network weights. It proposes a first-order approximation to estimate the gradients of architecture parameters, which is biased and may lead to the severe instability issue shown by [3]. Other works [38, 6] point out that architecture parameters will converge to a sharp local minimum resulting in the instability issue, so they introduce extra regularization items making architecture parameters converge to a flat local minimum.

---

*The SJTU authors are also with MoE Key Lab of Artificial Intelligence, SJTU. Junchi Yan is the corresponding author who is also with Shanghai AI Laboratory.

In this paper, we empirically show that the first-order approximation of optimal network weights sharpens the loss landscape and results in the instability issue of DARTS. It also shifts the global minimum, misleading the training of architecture parameters. To this end, we discard such approximation and turn to zero-order optimization algorithms, which can run without the requirement that the search loss is differentiable w.r.t. architecture parameters. Specifically, we introduce a novel NAS scheme named ZARTS, which outperforms DARTS by a large margin and can discover efficient architectures stably on multiple public benchmarks. This paper sheds light on the frontier of NAS by:

**1) Establishing zero-order based robust paradigm to solve bi-level optimization for NAS.** Differentiable architecture search has been a well-developed area [20, 34, 31] which solves the bi-level optimization of NAS by gradient descent algorithms. However, this paradigm suffers from the instability issue during search since biased approximation for optimal network weights distorts the loss landscape, as shown in Fig. 1 (a) and (b). To this end, we propose a flexible zero-order optimization NAS framework to solve the bi-level optimization problem, which is compatible with multiple potential gradient-free algorithms in the literature.

**2) Uncovering the connection between zero-order architecture search and DARTS.** This work introduces three representative zero-order optimization algorithms without enforcing the unverified differentiability assumption for search loss w.r.t. architecture parameters. We reveal the connections between the zero-order algorithms and gradient descent algorithm, showing that two implementations of ZARTS can be seen as gradient-free counterparts to DARTS, being more stable and robust.

**3) Strong empirical performance and robustness.** Experiments on four datasets and five search spaces show that, unlike DARTS, which suffers the severe instability issue [38, 3], ZARTS can stably discover effective architectures on various benchmarks. In particular, the searched architecture achieves 75.7% top-1 accuracy on ImageNet, outperforming DARTS and most of its variants. Also, our ZARTS can be further improved by combining with orthogonal DARTS variants, achieving 97.81% on CIFAR-10 after searching in 0.5 GPU-day.

## 2 Related Work

**One-shot Neural Architecture Search.** [1] constructs a supernet and all candidate architectures can be seen as its sub-graph. DARTS [20] introduces architecture parameters to represent the importance of operations in the supernet and update them by gradient descent. Some works [34, 31, 10] reduce the memory requirement in the search process. While [38, 6] point out the instability of DARTS, i.e., skip-connection gradually dominates the normal cells, leading to performance collapse during search.

**Bi-level Optimization for NAS.** NAS can be modeled as a bi-level optimization for architecture parameters and network weights. DARTS [20] proposes the first/second-order approximations to estimate gradients of architecture parameters so that they can be trained by gradient descent. However, we show that such an approximation will distort the loss landscape and mislead the training of architecture parameters. Amended-DARTS [3] derives an analytic formula of the gradients w.r.t. architecture parameters that require the Hessian inverse of network weights, which is even unfeasible to compute. This work discards the approximation in DARTS and attempts to solve the bi-level optimization by gradient-free algorithms.

**Zero-order Optimization.** Unlike gradient-based optimization methods that require the objective differentiable w.r.t. the parameters, zero-order optimization can train parameters when the gradient is unavailable or difficult to obtain, which has been widely used in adversarial robustness for neural networks [5, 16], meta learning [28], and transfer learning [29]. [22] aim at AutoML and utilize zero-order optimization to discover optimal configurations for ML pipelines. In this work (to our best knowledge), we make the first attempt to apply zero-order optimization to NAS and experiment with multiple algorithms, from vanilla random search [11] to more advanced and effective direct search [12], showing its great superiority against gradient-based methods.

## 3 Bi-level Optimization in DARTS

Following one-shot NAS [1], DARTS constructs a supernet stacked by normal and reduced cells. Cells in the supernet are denoted by directed acyclic graphs (DAG) with $N$ nodes $\{x_i\}_{i=1}^N$, which represents latent feature maps. Each edge $e_{i,j}$ contains multiple operations $\{o_{i,j}, o \in \mathcal{O}\}$, whose

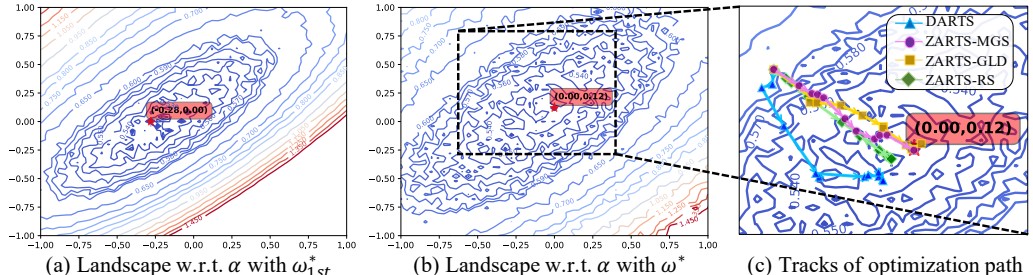

| (a) Landscape w.r.t. $\alpha$ with $\omega_{1st}^*$ | (b) Landscape w.r.t. $\alpha$ with $\omega^*$ | (c) Tracks of optimization path |

Figure 1: Loss landscapes w.r.t. architecture parameters $\alpha$ where the red star indicates the global minimum. (a) the landscape with $\omega_{1st}^*$. (b) the landscape with $\omega^*$, which is obtained by training $\omega$ for 10 iterations. To fairly compare the landscapes in (a) and (b), we utilize the same model and candidate $\alpha$ points. The first-order approximation sharpens the landscape. (c) displays the path of DARTS and ZARTS. Starting at the same initial point, ZARTS converges to the global minimum.

importance is represented by architecture parameters $\alpha_{i,j}^o$. Therefore, NAS can be modeled as a bi-level optimization problem by alternately updating the operation weights $\omega$ (parameters within candidate operations on each edge) and the architecture parameters $\alpha$:

$$\min_{\alpha} \ \mathcal{L}_{val}(\omega^*(\alpha), \alpha), \quad \text{s.t.} \ \omega^*(\alpha) = \arg\min_{\omega} \mathcal{L}_{train}(\omega, \alpha). \tag{1}$$

### 3.1 Fundamental Limitations in DARTS

By enforcing an unverified (and in fact difficult to verify) assumption that the search loss $\mathcal{L}_{val}(\omega^*(\alpha), \alpha)$ is differentiable w.r.t. $\alpha$, DARTS [20] proposes a second-order approximation for the optimal weights $\omega^*(\alpha)$ by applying one-step gradient descent:

$$\omega^*(\alpha) \approx \omega_{2nd}^*(\alpha) = \omega - \xi\nabla_{\omega}\mathcal{L}_{train}(\omega, \alpha) = \omega', \tag{2}$$

where $\xi$ is the learning rate to update network weights. Thus the gradient of the loss w.r.t. $\alpha$, $\nabla_{\alpha}\mathcal{L}_{val}(\omega^*(\alpha), \alpha)$, can be computed by the chain rule: $\nabla_{\alpha}\mathcal{L}_{val}(\omega^*(\alpha), \alpha) \approx \nabla_{\alpha}\mathcal{L}_{val}(\omega', \alpha) - \xi\nabla_{\alpha,\omega}^2\mathcal{L}_{train}(\omega, \alpha)\nabla_{\omega'}\mathcal{L}_{val}(\omega', \alpha)$. Nevertheless, the second-order partial derivative is hard to compute, so the authors adopt the difference method, which is proved in the appendix.

For efficiency, first-order approximation is introduced by assuming $\omega^*(\alpha)$ being independent of $\alpha$, as shown in Eq. 3, which is much faster and widely used in many variants of DARTS [7, 31, 38].

$$\omega^*(\alpha) \approx \omega_{1st}^*(\alpha) = w. \tag{3}$$

The gradient is then simplified as: $\nabla_{\alpha}\mathcal{L}_{val}(\omega^*(\alpha), \alpha) \approx \nabla_{\alpha}\mathcal{L}_{val}(\omega, \alpha)$, which exacerbates the estimation bias. Reexamining the definition of $\omega^*(\alpha)$ in Eq. 1, one would note that it is intractable to derive a mathematical expression for $\omega^*(\alpha)$, making $\mathcal{L}_{val}(\omega^*(\alpha), \alpha)$ even non-differentiable w.r.t. $\alpha$. Yet DARTS has to compromise with such approximations as Eq. 2 and Eq. 3 so that differentiability is established, and SGD can be applied. However, such sketchy estimation of optimal operation weights can distort the loss landscape w.r.t. architecture parameters and thus mislead the search procedure, shown in Fig. 1 and analyzed in the next section.

### 3.2 Distorted Landscape and Biased Optimization

Fig. 1 illustrates the loss landscape with perturbations on architecture parameters $\alpha$, showing how different approximations of $\omega^*$ affect the search process. We train a supernet for 50 epochs and randomly select two orthonormal vectors as the directions to perturb $\alpha$. The same group of perturbation directions is used to draw landscapes in Fig. 1(a) and (b) for a fair comparison. Fig. 1(a) shows the loss landscape with the first-order approximation in DARTS, $\omega_{1st}^*(\alpha) = \omega$, while Fig. 1(b) shows the loss landscape with more accurate $\omega^*(\alpha)$, which is obtained by fine-tuning the network weights $\omega$ for 10 iterations for each $\alpha$. Landscapes (contours) are plotted by evaluating $\mathcal{L}$ at grid points ranging from -1 to 1 at an interval of 0.02 in both directions. Global minima are marked with stars on the landscapes, from which we have two observations: 1) The approximation $\omega_{1st}^*(\alpha) = \omega$ shifts the global minimum and sharpens the landscape [2], which is the representative characteristic of

---

[2] A "sharp" landscape has denser contours than a "flat" one.

**Algorithm 1** ZARTS: Zero-order Optimization Framework for Architecture Search

---

**Hyper-parameters:** Operation weights $\boldsymbol{\omega}$, architecture parameters $\boldsymbol{\alpha}$, sampling number $N$, iteration number $M$, update estimation function $\phi(\cdot)$.
**repeat**
    **Sample candidates:** $\{\mathbf{u}_i\}_{i=1}^{N}$, and get $\boldsymbol{\alpha}_i^{\pm} = \boldsymbol{\alpha} \pm \mathbf{u}_i$. Estimate optimal operation weights $\boldsymbol{\omega}^*(\boldsymbol{\alpha}_i^{\pm})$ by descending $\nabla_{\boldsymbol{\omega}}\mathcal{L}_{train}(\boldsymbol{\omega}, \boldsymbol{\alpha}_i^{\pm})$ for $M$ iterations;
    **Get update direction:** $\mathbf{u}^* = \phi\left(\{\mathbf{u}_i, \boldsymbol{\omega}^*(\boldsymbol{\alpha}_i^{\pm})\}_{i=1}^{N}\right)$
    **Update architecture parameters:** $\boldsymbol{\alpha} \leftarrow \boldsymbol{\alpha} + \mathbf{u}^*$;
**until** Converged
$^\star$ The sampling strategies and update estimation functions $\phi(\cdot)$ for three different zero-order optimization algorithms are detailed in Table 1.

---

instability issue as pointed out by [38]. 2) Accurate estimation for $\boldsymbol{\omega}^*$ leads to a flatter landscape, indicating that the instability issue can be alleviated. Moreover, we display the landscape with second-order approximation $\boldsymbol{\omega}_{2nd}^*$ in the appendix, which is also sharp but slightly flatter than Fig. 1 (a). Consequently, we discard the first/second-order approximation in DARTS and instead use more accurate $\boldsymbol{\omega}^*$ coordinated with zero-order optimization.

Fig. 1 (c) shows the optimization paths of DARTS and three methods of ZARTS, illustrating how the approximation in DARTS affects the search process. Starting from the same random point, we update architecture parameters $\boldsymbol{\alpha}$ for ten iterations by DARTS and ZARTS, and draw the optimization path. ZARTS can gradually converge to the global minimum, while DARTS fails.

## 4 Zero-order Optimization for NAS

This paper goes beyond the 1st/2nd-order approximation in DARTS and proposes to train architecture parameters $\boldsymbol{\alpha}$ by zero-order optimization, allowing for more accurate estimation for $\boldsymbol{\omega}^*(\boldsymbol{\alpha})$. Alg. 1 outlines the generic form of our ZARTS framework. We adopt three representative techniques: a vanilla zero-order optimizer, random search (RS) [21], and two advanced algorithms: Maximum-likelihood Guided Parameter Search (MGS) [32] and GradientLess Descent (GLD) [12], presented in Sec. 4.1, 4.2, 4.3 as preliminaries. Further, we theoretically establish the connection between ZARTS and DARTS, showing that ZARTS with RS and MGS optimizer can be seen as an expansion of DARTS. In the following, we denote $\mathcal{L}(\boldsymbol{\alpha}) \triangleq \mathcal{L}_{val}(\boldsymbol{\omega}^*(\boldsymbol{\alpha}), \boldsymbol{\alpha})$ as the objective w.r.t. architecture parameters $\boldsymbol{\alpha} \in \mathbb{R}^d$ (Eq. 1), and $\mathcal{L}(\boldsymbol{\alpha} + \mathbf{u}) \triangleq \mathcal{L}_{val}(\boldsymbol{\omega}^*(\boldsymbol{\alpha} + \mathbf{u}), \boldsymbol{\alpha} + \mathbf{u})$, where $\mathbf{u}$ is the update for $\boldsymbol{\alpha}$.

### 4.1 Optimizer I: ZARTS-RS

We adopt *Multi-point estimator*, shown in Table 1, as a baseline of our ZARTS:

$$\hat{\nabla}_{\boldsymbol{\alpha}}\mathcal{L}(\boldsymbol{\alpha}) := \frac{\varphi(d)}{2\mu N} \sum_{i=1}^{N} \left[\mathcal{L}(\boldsymbol{\alpha} + \mu\mathbf{u}_i) - \mathcal{L}(\boldsymbol{\alpha} - \mu\mathbf{u}_i)\right]\mathbf{u}_i. \tag{4}$$

where $\mathbf{u} \sim q$ is sampled from a spherically symmetric distribution $q$, $\mu > 0$ is a smoothing parameter, and $\varphi(d)$ is a dimension-dependent factor related to $q$. Specifically, $\varphi(d) = 1$ when $q$ is a standard normal distribution $\mathcal{N}(\mathbf{0}, \mathbf{I})$, and $\varphi(d) = d$ when $q$ is a uniform distribution on a unit sphere $\mathbb{S}^{d-1}$.

### 4.2 Optimizer II: ZARTS-MGS

Maximum-likelihood guided parameter search (MGS) is an advanced zero-order optimization algorithm for machine translation [32]. We attempt to apply it to the NAS task. We first define a distribution for the update of architecture parameters, $\mathbf{u}$, as follows:

$$p(\mathbf{u}|\boldsymbol{\alpha}) = \frac{\widetilde{p}(\mathbf{u}|\boldsymbol{\alpha})}{Z(\boldsymbol{\alpha})} = \frac{1}{Z(\boldsymbol{\alpha})} \exp\left(-\frac{\mathcal{L}(\boldsymbol{\alpha} + \mathbf{u}) - \mathcal{L}(\boldsymbol{\alpha})}{\tau}\right), \tag{5}$$

where $\widetilde{p}(\mathbf{u}|\boldsymbol{\alpha}) = \exp\left(-[\mathcal{L}(\boldsymbol{\alpha} + \mathbf{u}) - \mathcal{L}(\boldsymbol{\alpha})]/\tau\right)$ is an unnormalized exponential distribution, and $Z(\boldsymbol{\alpha}) = \int \widetilde{p}(\mathbf{u}|\boldsymbol{\alpha})\mathrm{d}\mathbf{u}$ is its normalization coefficient. $\tau$ is a temperature parameter controlling the variance of the distribution.

Table 1: Configuration of three methods used in the ZARTS scheme. The main difference lies in the meaning of function $\phi(\cdot)$: RS follows the traditional gradient estimation algorithms, MGS estimates the update according to the improvement in the loss function, while GLD uses direct search. Note that the ZARTS framework is general and can support more configurations besides those listed.

| Algorithm | Sampling strategy | Update estimation function $\phi\left(\{\mathbf{u}_i, \boldsymbol{\omega}^*(\boldsymbol{\alpha}_i)\}_{i=1}^N\right)$ |
|---|---|---|
| ZARTS-RS | $\mathbf{u}_i \sim q(\mathbf{u}\|\boldsymbol{\alpha})$, any spherically symmetric distribution. | $\mathbf{u}^* = -\xi \cdot \frac{\varphi(d)}{2\mu N} \sum_{i=1}^N \left[\mathcal{L}(\boldsymbol{\alpha} + \mu \mathbf{u}_i) - \mathcal{L}(\boldsymbol{\alpha} - \mu \mathbf{u}_i)\right]\mathbf{u}_i$ (Eq. 4) |
| ZARTS-MGS | $\mathbf{u}_i \sim q(\mathbf{u}\|\boldsymbol{\alpha})$, any proposal distribution. | $\mathbf{u}^* = \sum_{i=1}^N \left[\frac{\widetilde{c}(\mathbf{u}_i\|\boldsymbol{\alpha})}{\sum_{j=1}^N \widetilde{c}(\mathbf{u_j}\|\boldsymbol{\alpha})}\mathbf{u}_i\right]$ (Eq. 8) |
| ZARTS-GLD | $\mathbf{u}_i \sim \mathbb{S}^{d-1}$, a uniform distribution on a unit sphere. | $\mathbf{u}^* = \arg\min_i \{\mathcal{L}(\hat{\boldsymbol{\alpha}})\|\hat{\boldsymbol{\alpha}} = \boldsymbol{\alpha}, \hat{\boldsymbol{\alpha}} = \boldsymbol{\alpha} + \mathbf{u}_i\}$ (Eq. 10) |

Intuitively, $\mathbf{u}$ with higher probability contributes more to the objective. Hence, the optimal update of architecture parameters can be estimated by $\mathbf{u}^* = \mathbb{E}_{\mathbf{u} \sim p(\mathbf{u}|\boldsymbol{\alpha})}[\mathbf{u}]$. However, since the probability $p(\mathbf{u}|\boldsymbol{\alpha})$ is an implicit function relying on $\mathcal{L}(\boldsymbol{\alpha} + \mathbf{u})$, making it impractical to obtain the expectation, we refer to [32] and apply importance sampling to sample from a proposal distribution $q(\mathbf{u}|\boldsymbol{\alpha})$ with known probability function:

$$\mathbf{u}^* = \mathbb{E}_{\mathbf{u} \sim p(\mathbf{u}|\boldsymbol{\alpha})}[\mathbf{u}] = \int \frac{\tilde{p}(\mathbf{u}|\boldsymbol{\alpha})}{Z(\boldsymbol{\alpha})}\mathbf{u}\mathrm{d}\mathbf{u} = \int q(\mathbf{u}|\boldsymbol{\alpha})\left[\frac{\tilde{p}(\mathbf{u}|\boldsymbol{\alpha})}{Z(\boldsymbol{\alpha})q(\mathbf{u}|\boldsymbol{\alpha})}\mathbf{u}\right]\mathrm{d}\mathbf{u}$$

$$= \mathbb{E}_{\mathbf{u} \sim q(\mathbf{u}|\boldsymbol{\alpha})}\left[\frac{\tilde{p}(\mathbf{u}|\boldsymbol{\alpha})}{Z(\boldsymbol{\alpha})q(\mathbf{u}|\boldsymbol{\alpha})}\mathbf{u}\right] \approx \frac{1}{N}\sum_{i=1}^N \left[\frac{\tilde{p}(\mathbf{u}_i|\boldsymbol{\alpha})}{Z(\boldsymbol{\alpha})q(\mathbf{u}_i|\boldsymbol{\alpha})}\mathbf{u}_i\right] \triangleq \hat{\mathbf{u}}^*, \tag{6}$$

where $\{\mathbf{u}_i\}_{i=1}^N$ are sampled from the proposal distribution $q(\mathbf{u}|\boldsymbol{\alpha})$. Similarly, the normalization coefficient $Z(\boldsymbol{\alpha})$ can be computed as follows:

$$Z(\boldsymbol{\alpha}) = \int \widetilde{p}(\mathbf{u}|\boldsymbol{\alpha})\mathrm{d}\mathbf{u} \approx \frac{1}{N}\sum_{i=1}^N \left[\frac{\tilde{p}(\mathbf{u}_i|\boldsymbol{\alpha})}{q(\mathbf{u}_i|\boldsymbol{\alpha})}\right]. \tag{7}$$

For convenience, we define a ratio representing the weight on each sample as $\widetilde{c}(\mathbf{u}|\boldsymbol{\alpha}) = \frac{\tilde{p}(\mathbf{u}|\boldsymbol{\alpha})}{q(\mathbf{u}|\boldsymbol{\alpha})}$. The optimal update for architecture parameters in Eq. 6 can be computed by:

$$\hat{\mathbf{u}}^* = \sum_{i=1}^N \left[\frac{\widetilde{c}(\mathbf{u}_i|\boldsymbol{\alpha})}{\sum_{j=1}^N \widetilde{c}(\mathbf{u}_j|\boldsymbol{\alpha})}\mathbf{u}_i\right] = \sum_{i=1}^N \left[\frac{\frac{\exp(-[\mathcal{L}(\boldsymbol{\alpha}+\mathbf{u}_i)-\mathcal{L}(\boldsymbol{\alpha})]/\tau)}{q(\mathbf{u}_i|\boldsymbol{\alpha})}}{\sum_{j=1}^N \frac{\exp(-[\mathcal{L}(\boldsymbol{\alpha}+\mathbf{u}_j)-\mathcal{L}(\boldsymbol{\alpha})]/\tau)}{q(\mathbf{u}_j|\boldsymbol{\alpha})}}\mathbf{u}_i\right]. \tag{8}$$

Finally, the architecture parameters are updated by $\boldsymbol{\alpha} \leftarrow \boldsymbol{\alpha} + \hat{\mathbf{u}}^*$. Importance sampling diagnostics are conducted to verify the effectiveness of ZARTS-MGS (in the appendix).

### 4.3 Optimizer III: ZARTS-GLD

Unlike the above two algorithms that estimate gradient or the update for $\boldsymbol{\alpha}$, [12] propose the so-called GradientLess Descent (GLD) algorithm, which falls into the category of truly gradient-free (or direct search) methods. The authors provide theoretical proof of the efficacy and efficiency of the GLD algorithm and suggestions on the choice of search radius boundaries. Specifically, they prove that the distance between the optimal minimum and the solution given by GLD is bounded and positively correlated with the condition number of the objective, where the condition number $Q$ is defined as:

$$Q = \max_{1 \leq i \leq K}\left\{\frac{|\mathcal{L}(\boldsymbol{\alpha} + \Delta_i) - \mathcal{L}(\boldsymbol{\alpha})| \cdot \|\boldsymbol{\alpha}\|}{\|\Delta_i\| \cdot |\mathcal{L}(\boldsymbol{\alpha})|}\right\}. \tag{9}$$

The loss landscape in Fig. 1(b) is pretty flat, implying a low condition number, thus the high efficiency of ZARTS-GLD. Specifically, at each iteration, with a predefined search radius boundary $[r, R]$, we independently sample candidate updates $\{\mathbf{u}_i\}$ for architecture parameters on spheres with various radii $\{2^{-k}R\}_{k=0}^{\log(R/r)}$ and perform function evaluation at these points. By comparing $\mathcal{L}(\boldsymbol{\alpha})$ and $\{\mathcal{L}(\boldsymbol{\alpha} + \mathbf{u}_i)\}$, $\boldsymbol{\alpha}$ steps to the point with minimum value, or stay at the current point if none of them makes an improvement. The architecture parameters are then updated by $\boldsymbol{\alpha} \leftarrow \boldsymbol{\alpha} + \mathbf{u}^*$.

$$\mathbf{u}^* = \arg\min_i\{\mathcal{L}(\hat{\boldsymbol{\alpha}})|\hat{\boldsymbol{\alpha}} = \boldsymbol{\alpha}, \hat{\boldsymbol{\alpha}} = \boldsymbol{\alpha} + \mathbf{u}_i\} \tag{10}$$

## 4.4 Connection between DARTS and ZARTS

The similarity between gradient-estimation-based zero-order optimization and SGD builds an essential connection when the objective function is differentiable. For ZARTS-RS, we adopt a multi-point estimator to approximate the gradient with a limited bound [23, 2]. Details are put in the supplementary material. Therefore, ZARTS-RS degenerates to second-order DARTS if $\mathcal{L}(\alpha)$ is indeed differentiable w.r.t $\alpha$ and the iteration number $M$ is set to 1.

Next, we theoretically show that MGS [32] degenerates to the gradient descent algorithm by using first-order Taylor approximation. Then we analyze the relation between ZARTS-MGS and DARTS.

**Proposition 1.** *When $\mathcal{L}(\alpha)$ in Eq. 1 is differentiable w.r.t. $\alpha$, MGS algorithm [32] degenerates to SGD (used in vanilla DARTS) by the first-order Taylor approximation for $\mathcal{L}(\alpha)$, i.e., $\mathbf{u}^* \propto -\nabla_\alpha \mathcal{L}(\alpha)$.*

*Proof.* Denote $\mathbf{g} \triangleq \nabla_\alpha \mathcal{L}(\alpha)$ as the gradient of $\mathcal{L}$. The Taylor series of $\mathcal{L}$ at $\alpha$ up to the first order gives $\mathcal{L}(\alpha + \mathbf{u}) - \mathcal{L}(\alpha) \approx \mathbf{u}^\top \mathbf{g}$. Applying it to $p(\mathbf{u}|\alpha)$ in Eq. 5 yields:

$$p(\mathbf{u}|\alpha) = \frac{e^{-\mathbf{u}^\top \mathbf{g}/\tau}}{Z(\mathbf{g})}, \ Z(\mathbf{g}) = \int_{\|\mathbf{u}\| \leq \varepsilon} e^{-\mathbf{u}^\top \mathbf{g}/\tau} \mathrm{d}\mathbf{u}. \tag{11}$$

Since $\|\mathbf{u}\|$ is constrained within $\varepsilon$ to make sure the rationality of first-order Taylor approximation, the optimal update $\mathbf{u}^*$ then becomes:

$$\mathbf{u}^* = \frac{\int_{\|\mathbf{u}\| \leq \varepsilon} \mathbf{u} \cdot e^{-\mathbf{u}^\top \mathbf{g}/\tau} \mathrm{d}\mathbf{u}}{Z(\mathbf{g})} = -\frac{\nabla_\mathbf{g} Z(\mathbf{g})}{\tau Z(\alpha, \mathbf{g})} = -\frac{1}{\tau} \nabla_\mathbf{g} \ln Z(\mathbf{g}). \tag{12}$$

Note that $\mathbf{u}^\top \mathbf{g} = -\|\mathbf{u}\|\|\mathbf{g}\| \cos \eta$, where $\eta$ is the angle between $\mathbf{u}$ and $\mathbf{g}$. According to the symmetry of integral, $Z(\mathbf{g})$ is determined once $\|\mathbf{g}\|$ is given. We formulate $Z(\mathbf{g})$ as $Z(\|\mathbf{g}\|)$ with the chain rule:

$$\mathbf{u}^* = -\frac{1}{\tau} \nabla_\mathbf{g} \ln Z(\mathbf{g}) = -\frac{1}{\tau} \nabla_\mathbf{g} \ln Z(\|\mathbf{g}\|) = -\frac{\nabla_{\|\mathbf{g}\|} Z(\|\mathbf{g}\|)}{\tau Z(\|\mathbf{g}\|)} \nabla_g \|\mathbf{g}\| = -\frac{\nabla_{\|\mathbf{g}\|} Z(\|\mathbf{g}\|)}{\tau Z(\|\mathbf{g}\|)\|\mathbf{g}\|} \mathbf{g}. \tag{13}$$

Since $Z(\|\mathbf{g}\|), \nabla_{\|\mathbf{g}\|} Z(\|\mathbf{g}\|), \|\mathbf{g}\|$ are all scalars, we have

$$\mathbf{u}^* \propto -\mathbf{g} = -\nabla_\alpha \mathcal{L}(\alpha) = -\nabla_\alpha \mathcal{L}_{val}(\omega^*(\alpha), \alpha). \tag{14}$$

That is, the optimal update $\mathbf{u}^*$ in MGS algorithm shares a common direction with the negative gradient $-\nabla_\alpha \mathcal{L}(\alpha)$, as used by gradient descent. $\qquad \square$

Based on Proposition 1, ZARTS-MGS can be seen as an expansion of DARTS, and it degrades to first-order and second-order DARTS when $\omega^*$ is estimated by $\omega^*_{1st}$ and $\omega^*_{2nd}$, respectively. In general, ZARTS-RS/-MGS can degenerate to DARTS, given the differentiability assumption.

However, unlike DARTS, which has to estimate $\omega^*(\alpha)$ by $\omega^*_{1st}$ or $\omega^*_{2nd}$ to satisfy the differentiablity property of $\mathcal{L}_{val}$ and update $\alpha$ by gradient descent algorithm, ZARTS, without such assumptions, can compute $\omega^*(\alpha)$ by training network weights $\omega$ for arbitrary numbers of iterations, leading to more robust and effective training for architecture parameters, as shown in the next section.

## 4.5 Variants and Speedup of ZARTS

ZARTS can be flexibly and seamlessly combined with variants of DARTS for boosting as it can be seen as an expansion of DARTS as analyzed in section 4.4. Here, we derive three variants of ZARTS by combining it with prior works: P-ZARTS based on P-DARTS [7], GZAS based on GDAS [10], and MergeZARTS based on MergeNAS [31]. Implementation details are illustrated in the appendix. These variants can speed up ZARTS twice and reduce the search cost to 0.5 GPU-day. Moreover, GZAS and MergeZARTS also reduce the GPU memory cost during the search process. The performance of these variants is reported in Table 5.

## 5 Experiments

We first verify the stability of ZARTS (with three zero-order optimization methods RS, MGS, GLD) on the four popular search spaces of R-DARTS [38] on three datasets including CIFAR-10 [17],

Table 3: Performance on CIFAR. The top reports the accuracy of the best model. The bottom gives the mean of four independent searches as used by [38, 6, 37]. $\diamond$ Reported by [10]. $\star$ by [38].

| | CIFAR-10 | Params (M) ↓ | Error (%) ↓ | Cost (GPU-days) ↓ | | CIFAR-100 | Params (M) ↓ | Error (%) ↓ | Cost (GPU-days) ↓ |
|---|---|---|---|---|---|---|---|---|---|
| best | DARTS (1st) [20] | 3.3 | 3.00 | 0.4 | best | AmoebaNet [26] | 3.1 | 18.93$^\diamond$ | 3150 |
| | DARTS (2nd) [20] | 3.4 | 2.76 | 1.0 | | PNAS [19] | 3.2 | 19.53$^\diamond$ | 150 |
| | P-DARTS [7] | 3.4 | 2.50 | 0.3 | | ENAS [25] | 4.6 | 19.43$^\diamond$ | 0.45 |
| | ISTA-NAS [36] | 3.3 | 2.54 | 0.05 | | P-DARTS [7] | 3.6 | 17.49 | 0.3 |
| | PR-DARTS [40] | 3.4 | 2.32 | 0.17 | | GDAS [10] | 3.4 | 18.38$^\diamond$ | 0.2 |
| | DARTS- [8] | 3.5 | 2.50 | 0.4 | | ROME [30] | 4.4 | 17.33 | 0.3 |
| | **ZARTS (ours)** | 3.5 | **2.46** | 1.0 | | PR-DARTS [40] | 3.4 | 16.45 | 0.17 |
| | | | | | | DARTS- [8] | 3.4 | 17.16 | 0.4 |
| | | | | | | **ZARTS (ours)** | 4.0 | **15.46** | 1.0 |
| average | DARTS(1st) [37] | - | 3.38±0.23 | 0.4 | average | DARTS [20] | - | 20.58±0.44$^\star$ | 0.4 |
| | SGAS (Cri.2) [18] | 3.9 | 2.67±0.21 | 0.3 | | R-DARTS [38] | - | 18.01±0.26$^\star$ | 1.6 |
| | R-DARTS [38] | - | 2.95±0.21 | 1.6 | | ROME [30] | 4.4 | 17.41±0.12 | 0.3 |
| | SDARTS-ADV [6] | 3.3 | 2.61±0.02 | 1.3 | | DARTS- [8] | 3.3 | 17.51±0.25 | 0.4 |
| | Amended-DARTS [3] | 3.3 | 2.71±0.09 | 1.7 | | **ZARTS (ours)** | 4.1±0.13 | **16.29±0.53** | 1.0 |
| | DARTS- [8] | 3.5±0.1 | 2.59±0.08 | 0.4 | | | | | |
| | **ZARTS (ours)** | 3.7±0.3 | **2.54±0.07** | 1.0 | | | | | |

CIFAR-100 [17], and SVHN [24]. We then follow Amended-DARTS [3] and empirically evaluate the convergence ability of our method by searching for 200 epochs. Performance trends of the discovered architectures are drawn in Fig. 2(a). Next, we compare with the peer methods on the search space of DARTS [20] to show the efficacy of our method. Finally, we derive three variants of ZARTS with faster search speed by combining ZARTS with the variants of DARTS, showing the flexibility and potential of our ZARTS framework. All the experiments are conducted on NVIDIA 2080Ti. The details of search spaces and experiment settings are given in the appendix.

## 5.1 Stability Evaluation

The instability of DARTS has drawn recent attention. Amended-DARTS [3] shows that skip-connection gradually dominates the discovered architectures after searching by DARTS for 200 epochs. R-DARTS [38] proposes four search spaces, S1-S4, which amplify the instability of DARTS, as such dominance occurs after only 50 epochs of searching. These studies expose the instability of gradient-based methods. To verify the stability of our method, we search on S1-S4 proposed by R-DARTS and conduct convergence analysis following Amended-DARTS.

**Performance on S1-S4.** We first search on the four spaces on CIFAR-10, CIFAR-100, and SVHN. The eval-

Table 2: Test error (%) with DARTS and its variants on S1-S4 search spaces, on CIFAR-10/100 and SVHN. We adopt the same settings as R-DARTS [39]. The best and second best are underlined in boldface and in boldface, respectively.

| | | DARTS | R-DARTS | | DARTS | | ZARTS (ours) | | |
|---|---|---|---|---|---|---|---|---|---|
| | | | DP | L2 | ES | ADA | RS | MGS | GLD |
| CIFAR-10 | S1 | 3.84 | 3.11 | 2.78 | 3.01 | 3.10 | 2.83 | **2.65** | **2.50** |
| | S2 | 4.85 | 3.48 | 3.31 | 3.26 | 3.35 | 3.35 | **3.24** | **3.08** |
| | S3 | 3.34 | 2.93 | **2.51** | 2.74 | 2.59 | 2.59 | **2.56** | 2.56 |
| | S4 | 7.20 | 3.58 | **3.56** | 3.71 | 4.84 | 4.90 | 3.70 | **3.52** |
| CIFAR-100 | S1 | 29.46 | 25.93 | 24.25 | 28.37 | 24.03 | 23.64 | **23.16** | 23.33 |
| | S2 | 26.05 | 22.30 | 22.24 | 23.25 | 23.52 | 21.54 | **20.91** | 21.13 |
| | S3 | 28.90 | 22.36 | 23.99 | 23.73 | 23.37 | 22.62 | **22.33** | **21.90** |
| | S4 | 22.85 | 22.18 | 21.94 | **21.26** | 23.20 | 23.33 | 21.31 | **21.00** |
| SVHN | S1 | 4.58 | 2.55 | 4.79 | 2.72 | 2.53 | **2.40** | 2.51 | **2.48** |
| | S2 | 3.53 | 2.52 | 2.51 | 2.60 | 2.54 | 2.52 | **2.45** | **2.48** |
| | S3 | 3.41 | 2.49 | 2.48 | 2.50 | 2.50 | **2.41** | 2.52 | **2.44** |
| | S4 | 3.05 | 2.61 | 2.50 | 2.51 | **2.46** | 2.59 | **2.48** | 2.53 |

uation settings are the same as R-DARTS [38]. Specifically, four tests are conducted on each benchmark, among which the best is reported in Table 2. We observe that ZARTS achieves outstanding performance with great robustness on 12 benchmarks. All three zero-order algorithms outperform DARTS notably. Even the vanilla zero-order algorithm ZARTS-RS achieves similar robust performance as R-DARTS, which verifies our analysis in Fig. 1, i.e., the coarse estimation $\omega_{1st}^*$ in DARTS distorts the landscape and causes instability. Though ZARTS-GLD performs best in Table 2, it falls into the category of direct search methods, requiring more sampling for candidate updates $\mathbf{u}$ and thus more search cost (2.2 GPU-days on the search space of DARTS) than ZARTS-MGS (1.0 GPU-days). ZARTS-MGS is chosen by default if not otherwise specified, for its cost-effectiveness.

**Convergence Analysis.** The convergence ability of NAS methods describes whether a search method can stably discover effective architectures along the search process, i.e., whether the discovered

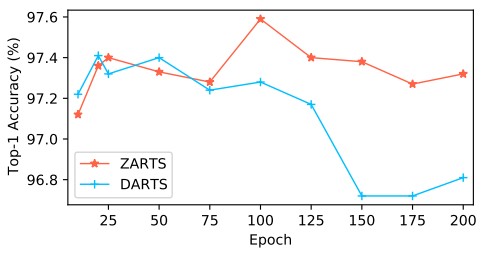

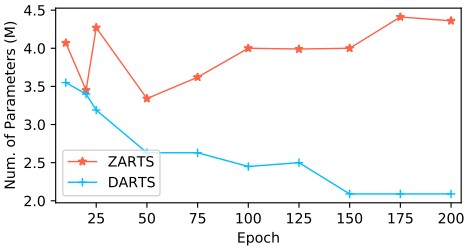

| (a) Trend of Top-1 accuracy | (b) Trend of Number of Parameters |

Figure 2: Trends of accuracy and model size in the search process of DARTS and ZARTS for 200 epochs on CIFAR-10. The top-1 accuracy is obtained by training models for 600 epochs.

architectures' ultimate performance (top-1 accuracy) can converge to a high value. Amended-DARTS [3] empirically shows that DARTS has a poor convergence ability: accuracy of the supernet increases but the ultimate performance of the searched network drops. Following Amended-DARTS, we run ZARTS and DARTS for 200 epochs and show the trend of performance and number of parameters in Fig. 2. Specifically, we derive one network every 25 epochs during the search process and train each network for 600 epochs to evaluate its ultimate performance. We observe that the networks discovered by ZARTS perform stably well (around 97.40% accuracy), while the performance of networks found by DARTS gradually drops. Moreover, the parameter number of networks searched by DARTS decreases significantly after 50 epochs, indicating that parameterless operations dominate the topology and the instability issue [39] occurs. On the contrary, ZARTS consistently discovers effective networks with about 4.0M parameters, showing the remarkable stability of our method. More details and visualization of architectures are shown in the appendix.

## 5.2 Performance on the Search Space of DARTS

**Results on CIFAR.** We conduct four parallel runs by searching with different random seeds and separately training the searched architectures for 600 epochs. The best and average accuracy of four parallel tests are reported in Table 3. In particular, ZARTS achieves 97.46% average accuracy and 97.54% best accuracy on CIFAR-10, outperforming DARTS and its variants. Also, compared with Amended-DARTS that approximates optimal operation weights $\omega^*(\alpha)$ by Hessian matrix, our method can stably discover effective architectures in fewer GPU days. Moreover, Table 3 shows our method

Table 4: Performance on ImageNet in DARTS's search space by two architectures. $^\dagger$ direct search on ImageNet.

| Models | FLOPs (M) ↓ | Params (M) ↓ | Top-1 Err. (%) ↓ | Cost (GPU-days) ↓ |
|---|---|---|---|---|
| AmoebaNet-A [26] | 555 | 5.1 | 25.5 | 3150 |
| NASNet-A [41] | 564 | 5.3 | 26.0 | 1800 |
| PNAS [19] | 588 | 5.1 | 25.8 | 225 |
| DARTS (2nd) [20] | 574 | 4.7 | 26.7 | 1.0 |
| P-DARTS [7] | 557 | 4.9 | 24.4 | 0.3 |
| PC-DARTS [35] | 586 | 5.3 | 25.1 | 0.1 |
| FairDARTS-B [9] | 541 | 4.8 | 24.9 | 0.4 |
| SNAS [33] | 522 | 4.3 | 27.3 | 1.5 |
| GDAS [10] | 581 | 5.3 | 26.0 | 0.2 |
| SPOS$^\dagger$ [13] | 323 | 3.5 | 25.6 | 12 |
| ProxylessNAS$^\dagger$ [4] | 465 | 7.1 | 24.9 | 8.3 |
| Amended-DARTS [3] | 586 | 5.2 | 24.7 | 1.7 |
| **ZARTS** (5.6M params) | 647 | 5.6 | **24.3** | 1.0 |
| **ZARTS** (5.0M params) | 573 | 5.0 | **24.5** | 1.0 |

achieves 83.71% and 84.54% for mean and best accuracy (i.e., 1- error%), on CIFAR-100, outperforming the compared methods by more than 1%.

**Results on ImageNet.** For the transferability test, we follow the settings of DARTS to transfer the network discovered on CIFAR-10 to ImageNet. Models are constructed by stacking 14 cells with 48 initial channels. We train 250 epochs with a batch size of 1024 by SGD with a momentum of 0.9 and a base learning rate of 0.5. We utilize the same data pre-processing strategies and auxiliary classifiers as DARTS. Table 4 shows the performance of our searched networks. ZARTS (5.6M) has 5.6M parameters and achieves 75.7% top-1 accuracy on the validation set of ImageNet, and ZARTS (5.0M) has 5.0M parameters and achieves 75.5% accuracy. Their structure details are given in the appendix, which has fewer skip connection operations than DARTS.

## 5.3 Variants and Speedup of ZARTS

The results of three variants of ZARTS, including the best and mean performance among four independent searches, are illustrated in Table 5. We search for 50 epochs by each variant and train the discovered architectures for 600 epochs. The search and evaluation settings are introduced in the appendix. Specifically, GZAS outperforms GDAS by 0.3% with a similar search cost, only 0.3 GPU-day. P-ZARTS outperforms P-DARTS by 0.15% and even surpasses ZARTS due to the handcrafted criteria in P-DARTS: setting the number of skip connection operations as 2. MergeZARTS achieves state-of-the-art performance with low search cost (0.5 GPU-day). We analyze that the weight merge technique among convolutions in MergeNAS reduces the

Table 5: We derive the variants of ZARTS by combining ZARTS with the variants of DARTS. Experiments are conducted on CIFAR-10. DARTS, GDAS, and P-DARTS report the best performance in their papers, and MergeNAS reports the average performance.

| Models | Params (M) ↓ | Error (%) ↓ | Cost (GPU-days) ↓ | Memory (GB) ↓ |
|---|---|---|---|---|
| DARTS (1st) [20] | 3.3 | 3.00 | 0.4 | 9.4 |
| DARTS (2nd) [20] | 3.4 | 2.76 | 1.0 | 9.4 |
| **ZARTS-best (ours)** | 3.5 | **2.46** | 1.0 | 9.4 |
| **ZARTS-avg. (ours)** | 3.7±0.3 | **2.54±0.07** | 1.0 | 9.4 |
| GDAS [10] | 3.4 | 2.93 | 0.2 | 3.1 |
| **GZAS-best (ours)** | 3.7 | **2.58** | 0.3 | 3.1 |
| **GZAS-avg. (ours)** | 3.5±0.2 | **2.66±0.07** | 0.3 | 3.1 |
| P-DARTS [7] | 3.4 | 2.50 | 0.3 | 9.4 |
| **P-ZARTS-best (ours)** | 3.5 | **2.30** | 0.4 | 9.4 |
| **P-ZARTS-avg. (ours)** | 3.3±0.2 | **2.41±0.15** | 0.4 | 9.4 |
| MergeNAS (1st) [31] | 2.9 | 2.73±0.02 | 0.2 | 4.4 |
| MergeNAS (2nd) [31] | 2.9 | 2.68±0.01 | 0.6 | 4.4 |
| **MergeZARTS-best (ours)** | 4.0 | **2.19** | 0.5 | 4.4 |
| **MergeZARTS-avg. (ours)** | 3.8±0.2 | **2.36±0.18** | 0.5 | 4.4 |

redundant network weights $\omega$, which alleviates the training difficulty for supernet and thus makes $\omega^*(\alpha)$ more accurate after $M$ iterations of gradient descent.

## 6 Conclusion and Future Work

DARTS has been a dominant paradigm in NAS, while its instability issue has received increasing attention [3, 38, 6]. In this work, we have empirically shown that the instability issue results from the first-order approximation for optimal network weights and the optimization gap in DARTS, which is also raised in the recent study [3]. To step out of such a bottleneck, this work proposes a robust search framework named ZARTS, allowing for higher-order approximation for $\omega^*(\alpha)$ and supporting multiple combinations of zero-order optimization algorithms. Specifically, we adopt three representative methods for experiments and reveal the connection between ZARTS and DARTS. Extensive experiments on various benchmarks show the effectiveness and robustness of ZARTS. To our best knowledge, this is the first work that applies zero-order optimization to one-shot NAS, providing a promising paradigm to solve the bi-level optimization problem for NAS.

**Limitation & future work:** This work is limited in directly adopting off-the-shelf zero-order solvers. There are potential directions for future work: i) We adopt three existing zero-order solvers in Table 1, which also suggests new solvers may also be readily reused to improve ZARTS. In contrast, this feature is not allowed in DARTS as there is little option for the gradient-descent solver. ii) Our experiments focus on classification tasks, while ZARTS can also search network architectures for more complex tasks. We derive three ZARTS variants from DARTS variants to speed up the search process. GZAS and MergeZARTS can also reduce the GPU memory requirements, making it possible to search on more complex search spaces or tasks, e.g., object detection and segmentation. iii) ZARTS supports searching for the targets whose gradients are intractable or hard to obtain, e.g., FLOPs and latency, since ZARTS discards the differentiability assumption and leverages gradient-free algorithms.

**Potential negative social impact:** Our automatic approach can save many human efforts which may cause job lost in industry. The technology may also be abused by people who may create evil AI.

## Acknowledgement

This work was partly supported by National Key Research and Development Program of China (2020AAA0107600), National Natural Science Foundation of China (61972250, 72061127003), and Shanghai Municipal Science and Technology (Major) Project (22511105100, 2021SHZDZX0102).

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
