# Supplementary Materials for "ZARTS: On Zero-order Optimization for Neural Architecture Search"

## 1 Appendix

### 1.1 Estimation for Second-order Partial Derivative in DARTS

[10] introduce second-order approximation to estimate optimal network weights, i.e., $\boldsymbol{\omega}^* \approx \boldsymbol{\omega}' = \boldsymbol{\omega} - \xi\nabla_{\boldsymbol{\omega}}\mathcal{L}_{train}(\boldsymbol{\omega}, \boldsymbol{\alpha})$, so that $\nabla_{\boldsymbol{\alpha}}\mathcal{L}_{val}(\boldsymbol{\omega}^*(\boldsymbol{\alpha}), \boldsymbol{\alpha}) \approx \nabla_{\boldsymbol{\alpha}}\mathcal{L}_{val}(\boldsymbol{\omega}', \boldsymbol{\alpha}) - \xi\nabla^2_{\boldsymbol{\alpha},\boldsymbol{\omega}}\mathcal{L}_{train}(\boldsymbol{\omega}, \boldsymbol{\alpha})\nabla_{\boldsymbol{\omega}'}\mathcal{L}_{val}(\boldsymbol{\omega}', \boldsymbol{\alpha})$. However, the second-order partial derivative is hard to compute, so the authors estimate it as follows:

$$\nabla^2_{\boldsymbol{\alpha},\boldsymbol{\omega}}\mathcal{L}_{train}(\boldsymbol{\omega}, \boldsymbol{\alpha})\nabla_{\boldsymbol{\omega}'}\mathcal{L}_{val}(\boldsymbol{\omega}', \boldsymbol{\alpha}) \approx \frac{\nabla_{\boldsymbol{\alpha}}\mathcal{L}_{train}(\boldsymbol{\omega}^+, \boldsymbol{\alpha}) - \nabla_{\boldsymbol{\alpha}}\mathcal{L}_{train}(\boldsymbol{\omega}^-, \boldsymbol{\alpha})}{2\epsilon}, \quad (1)$$

where $\boldsymbol{\omega}^{\pm} = \boldsymbol{\omega} \pm \epsilon\nabla_{\boldsymbol{\omega}'}\mathcal{L}_{val}(\boldsymbol{\omega}', \boldsymbol{\alpha})$, and $\epsilon = \frac{0.01}{\|\nabla_{\boldsymbol{\omega}'}\mathcal{L}_{val}(\boldsymbol{\omega}', \boldsymbol{\alpha})\|_2}$. Here, we prove that the above approximation in Eq. 1 is difference method.

*Proof.* First of all, to simplify the writing, we make the following definitions:

$$f(\boldsymbol{\omega}, \boldsymbol{\alpha}) = \nabla_{\boldsymbol{\alpha}}\mathcal{L}_{train}(\boldsymbol{\omega}, \boldsymbol{\alpha}), \quad g(\boldsymbol{\omega}, \boldsymbol{\alpha}) = \mathcal{L}_{val}(\boldsymbol{\omega}, \boldsymbol{\alpha}). \quad (2)$$

Then the left term in Eq. 1 can be simplified as:

$$\nabla^2_{\boldsymbol{\alpha},\boldsymbol{\omega}}\mathcal{L}_{train}(\boldsymbol{\omega}, \boldsymbol{\alpha})\nabla_{\boldsymbol{\omega}'}\mathcal{L}_{val}(\boldsymbol{\omega}', \boldsymbol{\alpha}) = \nabla_{\boldsymbol{\omega}}f(\boldsymbol{\omega}, \boldsymbol{\alpha}) \cdot \nabla_{\boldsymbol{\omega}'}g(\boldsymbol{\omega}', \boldsymbol{\alpha}) \quad (3)$$

$$= \nabla_{\boldsymbol{\omega}}f(\boldsymbol{\omega}, \boldsymbol{\alpha}) \cdot \frac{\nabla_{\boldsymbol{\omega}'}g(\boldsymbol{\omega}', \boldsymbol{\alpha})}{\|\nabla_{\boldsymbol{\omega}'}g(\boldsymbol{\omega}', \boldsymbol{\alpha})\|_2} \cdot \|\nabla_{\boldsymbol{\omega}'}g(\boldsymbol{\omega}', \boldsymbol{\alpha})\|_2 = \nabla_{\boldsymbol{\omega}}f(\boldsymbol{\omega}, \boldsymbol{\alpha}) \cdot \boldsymbol{l} \cdot \|\nabla_{\boldsymbol{\omega}'}g(\boldsymbol{\omega}', \boldsymbol{\alpha})\|_2, \quad (4)$$

where $\boldsymbol{l} = \frac{\nabla_{\boldsymbol{\omega}'}g(\boldsymbol{\omega}', \boldsymbol{\alpha})}{\|\nabla_{\boldsymbol{\omega}'}g(\boldsymbol{\omega}', \boldsymbol{\alpha})\|_2}$ is a unit vector. We notice $\nabla_{\boldsymbol{\omega}}f(\boldsymbol{\omega}, \boldsymbol{\alpha}) \cdot \boldsymbol{l}$ is the directional derivative of $f(\boldsymbol{\omega}, \boldsymbol{\alpha})$ along direction $\boldsymbol{l}$, which can be estimated by difference method with a small perturbation $\epsilon' = 0.01$:

$$\nabla_{\boldsymbol{\omega}}f(\boldsymbol{\omega}, \boldsymbol{\alpha}) \cdot \boldsymbol{l} \cdot \|\nabla_{\boldsymbol{\omega}'}g(\boldsymbol{\omega}', \boldsymbol{\alpha})\|_2 \approx \frac{f(\boldsymbol{\omega} + \epsilon'\boldsymbol{l}, \boldsymbol{\alpha}) - f(\boldsymbol{\omega} - \epsilon'\boldsymbol{l}, \boldsymbol{\alpha})}{2\epsilon'} \cdot \|\nabla_{\boldsymbol{\omega}'}g(\boldsymbol{\omega}', \boldsymbol{\alpha})\|_2 \quad (5)$$

Moreover, we define $\epsilon = \frac{\epsilon'}{\|\nabla_{\boldsymbol{\omega}'}g(\boldsymbol{\omega}', \boldsymbol{\alpha})\|_2}$. Then $\boldsymbol{\omega} \pm \epsilon'\boldsymbol{l} = \boldsymbol{\omega} \pm \epsilon\nabla_{\boldsymbol{\omega}'}g(\boldsymbol{\omega}', \boldsymbol{\alpha}) \triangleq \boldsymbol{\omega}^{\pm}$, so Eq. 5 can be simplified as:

$$\frac{f(\boldsymbol{\omega} + \epsilon'\boldsymbol{l}, \boldsymbol{\alpha}) - f(\boldsymbol{\omega} - \epsilon'\boldsymbol{l}, \boldsymbol{\alpha})}{2\epsilon'} \cdot \|\nabla_{\boldsymbol{\omega}'}g(\boldsymbol{\omega}', \boldsymbol{\alpha})\|_2 = \frac{f(\boldsymbol{\omega}^+, \boldsymbol{\alpha}) - f(\boldsymbol{\omega}^-, \boldsymbol{\alpha})}{2\epsilon}. \quad (6)$$

Substituting $f(\boldsymbol{\omega}, \boldsymbol{\alpha})$ in Eq. 2 with Eq. 6 results in Eq. 1. Therefore second-order approximation in DARTS utilizes difference method, which is also a zero-order optimization algorithm. □

### 1.2 Loss Landscape w.r.t. Architecture Parameters

To draw loss landscapes w.r.t. $\boldsymbol{\alpha}$, we train a supernet for 50 epochs and randomly select two orthonormal vectors as the directions to perturb $\boldsymbol{\alpha}$. The same group of perturbation directions is used to draw landscapes for a fair comparison. Landscapes are plotted by evaluating $\mathcal{L}$ at grid points ranged from -1 to 1 at an interval of 0.02 in both directions. Fig. 1 illustrates landscapes (contours)

36th Conference on Neural Information Processing Systems (NeurIPS 2022).

w.r.t. $\boldsymbol{\alpha}$ under different order of approximation for optimal network weights, showing that both first- and second-order approximation sharpen the landscape and in turn lead to incorrect global minimum. In this work, we obtain $\boldsymbol{\omega}^*(\boldsymbol{\alpha})$ by fixing $\boldsymbol{\alpha}$ and fine-tuning network weights for $M$ iterations (Fig. 1 (c)). Selection of $M$ is also analyzed in Appendix 2.3, showing that $M = 10$ iterations is accurate enough to estimate optimal network weights.

## 1.3 Details of ZARTS-MGS Algorithm

**Selection of the Proposal Distribution $q(\mathbf{u}|\boldsymbol{\alpha})$.** Since the probability function of distribution $p$ is intractable, we sample from a proposal distribution $q$ and approximate the optimal update of architecture parameters by Eq. 7.

$$\hat{\mathbf{u}}^* = \sum_{i=1}^N \left[ \frac{\widetilde{c}(\mathbf{u}_i|\boldsymbol{\alpha})}{\sum_{j=1}^N \widetilde{c}(\mathbf{u}_j|\boldsymbol{\alpha})} \mathbf{u}_i \right] = \sum_{i=1}^N \left[ \frac{\frac{\exp(-[\mathcal{L}(\boldsymbol{\alpha}+\mathbf{u}_i)-\mathcal{L}(\boldsymbol{\alpha})]/\tau)}{q(\mathbf{u}_i|\boldsymbol{\alpha})}}{\sum_{j=1}^N \frac{\exp(-[\mathcal{L}(\boldsymbol{\alpha}+\mathbf{u}_j)-\mathcal{L}(\boldsymbol{\alpha})]/\tau)}{q(\mathbf{u}_j|\boldsymbol{\alpha})}} \mathbf{u}_i \right]. \tag{7}$$

The proposal distribution $q$ affects the efficiency of sampling. Specifically, an ideal $q$ should be as close to $p$ as possible when the sampling number is limited. Following [14], we set the proposal distribution $q$ to a mixture of two Gaussian distributions, one of which is centered at the negative gradient of the loss function with current weights:

$$q(\mathbf{u}|\boldsymbol{\alpha}) = (1-\lambda)\mathcal{N}(-\nabla_{\boldsymbol{\alpha}}\mathcal{L}_{val}(\boldsymbol{\omega},\boldsymbol{\alpha}), \sigma^2) + \lambda\mathcal{N}(0, \sigma^2), \tag{8}$$

where $\sigma$ is the standard deviation. Intuitively, first-order DARTS [10] gives a hint: it updates the architecture parameters $\boldsymbol{\alpha}$ in the direction of $-\nabla_{\boldsymbol{\alpha}}\mathcal{L}_{val}(\boldsymbol{\omega},\boldsymbol{\alpha})$. The gradient is an imperfect but workable direction with easy access.

**Importance Sampling Diagnostics.** To demonstrate that importance sampling and our choice of the proposal distribution is indeed appropriate in our case, we evaluate the effectiveness of $q$ (the proposal distribution) quantitatively by the following experiments. According to [12], effective sample size (ESS) $N_e$ is a popular indicator defined as $N_e = \frac{1}{\sum_{i=1}^N c_i^2}$. For $N$ samples, a larger $N_e \in [1, N]$

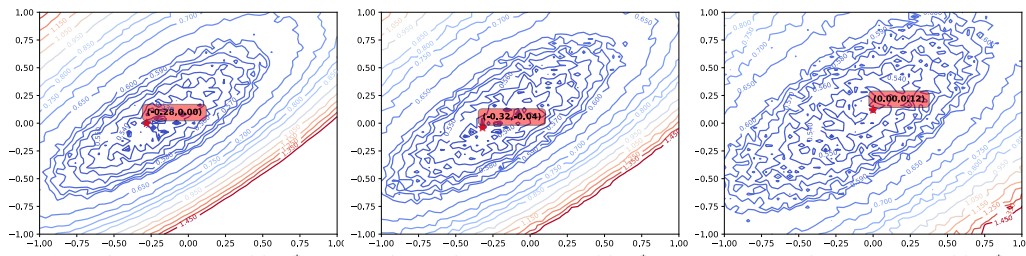

(a) Landscape w.r.t. $\alpha$ with $\omega_{1st}^*$     (b) Landscape w.r.t. $\alpha$ with $\omega_{2nd}^*$     (c) Landscape w.r.t. $\alpha$ with $\omega^*$

Figure 1: Loss landscapes w.r.t. architecture parameters $\boldsymbol{\alpha}$. In (a), we illustrate the landscape with first-order approximation. In (b), we illustrate the landscape with second-order approximation. In (c), we obtain $\boldsymbol{\omega}^*$ by training network weights $\boldsymbol{\omega}$ for 10 iterations, and illustrate the landscape w.r.t. $\boldsymbol{\alpha}$ with $\boldsymbol{\omega}^*$. To fairly compare the landscapes, we utilize the same model and candidate $\boldsymbol{\alpha}$ points. We observe the first/second-order approximations both sharpen the landscape.

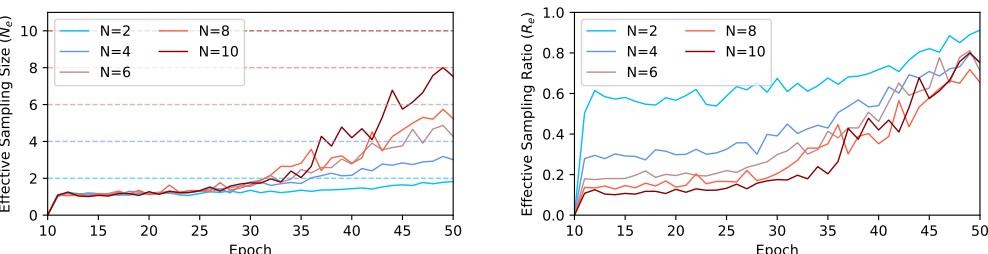

(a) ESS $N_e$ versus epochs over sampling number $N$.    (b) ESR $R_e$ versus epochs over sampling number $N$.

Figure 2: ESS $N_e$ and ESR $R_e$ versus epochs with different sampling numbers $N$ in the search stage on CIFAR-10. We fix iteration number $M = 10$ for all settings.

usually indicates a more effective sampling. On the contrary, small $N_e$ implies imbalanced sample weights and therefore is unreliable [12].

To evaluate the effectiveness of sampling in ZARTS-MGS, we set the sampling number $N$ to various values and plot $N_e$ versus epochs in each case. As is shown in Fig. 2(a), $N_e$ gradually approaches $N$ in all settings, indicating that different sampling numbers in our setting are meaningful, including larger ones (otherwise $N_e$ may "saturate").

As a further exploration, we define effective sample ratio (ESR) $R_e$ as the ratio of effective sampling to all samples:

$$R_e = \frac{N_e}{N}. \tag{9}$$

The value of $R_e$ denotes the bias between the target distribution $p$ and proposal distribution $q$, and a smaller value indicates a greater difference. $R_e$ is plotted against epochs for various $N$ in Fig. 2(b). On the one hand, $R_e$ at epoch 50 stabilizes at 0.7 as $N$ increases, which is an acceptable level of bias between $p$ and $q$ and supports our choice of $q$; on the other hand, we notice $R_e$ has already converged when $N \geq 4$. Considering the trade-off between estimation accuracy and speed, we set $N = 4$ as default, which is further discussed in Appendix 2.3.

## 2   Supplementary Experiments

### 2.1   Details of Search Spaces

**DARTS's Standard Search Space.** The operation set $\mathcal{O}$ contains 7 basic operations: skip connection, max pooling, average pooling, $3 \times 3$ separable convolution, $5 \times 5$ separable convolution, $3 \times 3$ dilated separable convolution, and $5 \times 5$ dilated separable convolution. Though zero operation is included in the origin search space of DARTS [10], it will never be selected in the searched architecture. Therefore, we remove zero operation from the search space. We search and evaluate on the CIFAR-10 [9] dataset in this search space, and then transfer the searched model to ImageNet [6]. Additionally, in our convergence analysis, we search by DARTS and ZARTS in this search space for 200 epochs.

**RDARTS's Search Spaces S1-S4.** To evaluate the stability of search algorithm, RDARTS [17] designs four search spaces where DARTS suffers from instability severely, i.e. normal cells are dominated by parameter-less operations (such as identity and max pooling) after searching for 50 epochs. In S1, each edge in the supernet only has two candidate operations, but the candidate operation set for each edge differs; in S2, the operation set $\mathcal{O}$ only contains $3 \times 3$ separable convolution and identity for all edges; in S3, $\mathcal{O}$ contains $3 \times 3$ separable convolution, identity, and zero for all edges; in S4, $\mathcal{O}$ contains $3 \times 3$ separable convolution and noise operation for all edges. Please refer to [17] for more details of the search spaces.

### 2.2   Experiment Settings

**Search Settings.** Similar to DARTS, we construct a supernet by stacking 8 cells with 16 initial channels. We apply Alg. 1 to train architecture parameters $\boldsymbol{\alpha}$ for 50 epochs. Two hyper-parameters of ZARTS, sampling number $N$ and iteration number $M$, are set to 4 and 10 respectively. Ablation studies of the two hyper-parameters are analyzed in Appendix 2.3. The setup for training $\boldsymbol{\omega}$ follows DARTS: SGD optimizer with a momentum of 0.9 and a base learning rate of 0.025. Our experiments are conducted on NVIDIA 2080Ti. ZARTS-MGS algorithm is used in supplementary experiments by default.

**Evaluation Settings.** We follow DARTS [10] and construct models by stacking 20 cells with 36 initial channels. Models are trained for 600 epochs by SGD with a batch size of 96. Cutout and auxiliary classifiers are used as introduced by DARTS.

### 2.3   Ablation Studies

There are two hyper-parameters in our method: sampling number $N$ and iteration number $M$. As introduced in Section 4, $N$ samples of update step of architecture parameters $\mathbf{u}_i$ are drawn to estimate the optimal update $\mathbf{u}^*$. For each sampled $\mathbf{u}_i$, we approximate the optimal weights $\boldsymbol{\omega}^*(\boldsymbol{\alpha} + \mathbf{u}_i)$ for

each sample by training $\omega$ for $M$ iterations. To evaluate the sensitivity of our method to the two hyper-parameters above, we conduct ablation studies on the standard search space of DARTS on CIFAR-10 dataset.

**Sensitivity to the Sampling Number $N$.** For various sampling numbers $N$, the average performance of three parallel searches with different random seeds is reported in Table 1 (left). In this experiment, iteration number $M$ is fixed to 10. When $N = 2$, ZARTS achieves 97.37% accuracy with 2.87M parameters. The number of parameters of searched network increases as $N$ increases, and the performance of searched network gets stable when $N \geq 4$. Our method performs better than DARTS when $N = 4$, with similar search cost (1.0 GPU days). When $N = 6$, ZARTS achieves its best accuracy (97.49%) and costs 1.1 GPU days. When $N$ continues to increase, our method attempts to find more complex architectures (with 4.31M and 4.26M parameters).

**Sensitivity to the Iteration Number $M$.** DARTS and its variants [16, 18] assume that the optimal operation weights $\omega^*(\alpha)$ is differentiable w.r.t. architecture parameters $\alpha$, which has not been theoretically proved. In this work, we relax the above assumption and adopt zero-order optimization to update $\alpha$. As introduced in Section 4, we perform multiple iterations of gradient descent on operation weights to accurately estimate $\omega^*(\alpha)$. To further confirm our analysis on the impact of iteration number $M$, we search with various values of $M$ and report the average performance of three parallel searches with different random seeds in Table 1 (right). In this experiment, we set the sampling number $N$ to 4. The results reveal that the performance of searched model improves as $M$ increases and the highest accuracy is achieved at $M = 10$, which supports our analysis that inaccurate estimation for optimal operation weights $\omega^*(\alpha)$ can mislead the search procedure.

## 2.4 Comparision with Peer Methods on S1-S4

Unlike R-DARTS [18] that constructs models by stacking 8 cells and 16 inital channels, SDARTS [3] builds models by stacking 20 cells and 36 initial channels. To compare with SDARTS for fair, we follow its settings and report our results in Table 2. Specifically, we conduct four parallel tests on each benchmark by searching with different random seeds. Tabel 2 reports the best and average performance of our method. Note that other methods in Tabel 2 only report the best performance of four parallel tests. According to the results, we observe our ZARTS achieves state-of-the-art on 7 benchmarks and SDARTS-ADV slightly outperforms our ZARTS on 5 benchmarks.

## 2.5 Convergence Analysis

The convergence ability of NAS methods describes whether a search method can stably discover effective architectures along the search process. A robust and effective NAS method should be able to converge to exemplary architectures with high performance. This work follows Amended-DARTS [1] and evaluates the convergence ability by searching for an extended period (200 epochs). However, since it is time-consuming to train every derived architecture along the search process, we illustrate the trend of number of parameterless operations (pooling and identity operations) in each normal cell to represent the performance of architectures (Fig. 3). Recent works [4, 1, 18] show that architecture with more than 4 parameterless operations (especially identity operations) usually has a bad performance, which is a typical phenomenon of the instability issue. Here, we show the number of parameterless operations of our ZARTS in Fig. 3 and compare with another three methods, including DARTS, PC-DARTS and S-DARTS. We observe that the architectures discovered by DARTS, PC-DARTS and S-DARTS will be gradually dominated by parameterless operations

Table 1: Comparison of different sampling numbers *(left)* and iteration numbers *(right)* to approximate the optimal update for architecture parameters on the standard search space of DARTS on CIFAR-10 dataset. For each setting, three parallel tests are conducted by searching on different random seeds and the mean and standard deviation of top-1 accuracy are reported.

| Sampling number | $N$=2 | $N$=4 | $N$=6 | $N$=8 | Iteration number | $M$=2 | $M$=5 | $M$=8 | $M$=10 |
|---|---|---|---|---|---|---|---|---|---|
| Error (%) | 2.63 | 2.54 | 2.51 | 2.57 | Error (%) | 2.62 | 2.60 | 2.57 | 2.54 |
| STD | ±0.12 | ±0.07 | ±0.09 | ±0.11 | STD | ±0.15 | ±0.09 | ±0.03 | ±0.07 |
| Params (M) | 2.87 | 3.71 | 3.53 | 4.31 | Params (M) | 2.91 | 3.40 | 3.52 | 3.71 |
| Cost (GPU days) | 0.5 | 1.0 | 1.1 | 1.5 | Cost (GPU-days) | 0.3 | 0.6 | 0.8 | 1.0 |

Table 2: Comparison with peer methods under the settings of SDARTS. (*left*) Test error of other methods are obtained from SDARTS [3], indicating the best performance among four replicate experiments with different random seeds. Note that 'RS' in SDARTS indicates random smoothing technique, while 'RS' in ZARTS indicates random search, a zero-order optimization algorithm. The best and second best is underlined in boldface and in boldface, respectively. (*right*) We report the average error and standard deviation of our method among four replicate experiments. Since PC-DARTS and SDARTS do not provide the average performance, we only compare the three implementations of our ZARTS.

| | | PC-DARTS | SDARTS RS | SDARTS ADV | ZARTS RS | ZARTS MGS | ZARTS GLD | | | ZARTS (avg.±std) RS | ZARTS (avg.±std) MGS | ZARTS (avg.±std) GLD |
|---|---|---|---|---|---|---|---|---|---|---|---|---|
| CIFAR-10 | S1 | 3.11 | 2.78 | 2.73 | 2.83 | **2.65** | **2.50** | CIFAR-10 | S1 | 3.10±0.20 | 2.79±0.14 | **2.73±0.07** |
| | S2 | 3.02 | 2.75 | 2.65 | **2.41** | **2.39** | 2.60 | | S2 | **2.60±0.14** | 2.65±0.17 | 2.67±0.08 |
| | S3 | **2.51** | 2.53 | **2.49** | 2.59 | 2.56 | 2.56 | | S3 | 2.89±0.30 | 2.74±0.10 | **2.70±0.09** |
| | S4 | 3.02 | 2.93 | 2.87 | 3.35 | **2.74** | **2.63** | | S4 | 4.11±1.07 | **2.99±0.21** | 3.35±0.93 |
| CIFAR-100 | S1 | 18.87 | 17.02 | **16.88** | **17.38** | 17.62 | 17.40 | CIFAR-100 | S1 | 18.07±0.55 | 18.20±0.48 | **17.83±0.44** |
| | S2 | 18.23 | 17.56 | 17.24 | **16.05** | **16.41** | 16.69 | | S2 | **17.04±0.70** | 17.35±0.75 | 17.14±0.39 |
| | S3 | 18.05 | 17.73 | 17.12 | 17.22 | **17.03** | **16.58** | | S3 | 18.14±0.72 | 17.72±0.46 | **16.99±0.38** |
| | S4 | 17.16 | 17.17 | **15.46** | 18.23 | 16.57 | **15.97** | | S4 | 19.08±1.01 | 17.33±0.73 | **16.63±0.75** |
| SVHN | S1 | 2.28 | 2.26 | 2.16 | **2.09** | **2.13** | 2.14 | SVHN | S1 | 2.21±0.10 | **2.17±0.03** | 2.20±0.04 |
| | S2 | 2.39 | 2.37 | 2.07 | **2.06** | **2.06** | 2.15 | | S2 | 2.16±0.10 | **2.10±0.03** | 2.22±0.07 |
| | S3 | 2.27 | 2.21 | **2.05** | 2.17 | 2.20 | **2.07** | | S3 | 2.27±0.14 | 2.25±0.05 | **2.13±0.07** |
| | S4 | 2.37 | 2.35 | **1.98** | 2.49 | **2.04** | 2.15 | | S4 | 2.56±0.09 | **2.20±0.11** | 2.26±0.09 |

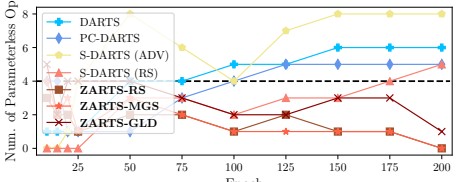

(a) Trend of Number of Parameterless Operations

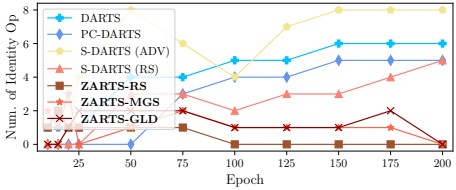

(b) Trend of Number of Identity Operations

Figure 3: Trends of number of parameterless operations and identity operations in each normal cell searched by different NAS methods on CIFAR-10 for 200 epochs. The parameterless operations include max pooling, average pooling, and identity operation.

(especially identity operation), implying that the instability issue occurs. In contrast, our ZARTS can stably control the number of parameterless operations.

## 2.6 Implementation Details of the Variants of ZARTS

Based on the variants of DARTS, we derive three variants of ZARTS: GZAS based on GDAS [7], P-ZARTS based on P-DARTS [4], and MergeZARTS based on MergeNAS [13]. We adopt MGS as the default zero-order optimizer. Here, we introduce the implementation details of these three variants.

**GZAS:** We sample and activate only one candidate operation for each super-edge using the Gumbel reparameterization technique. Since only a sub-network of the supernet is activated, GZAS has low GPU memory requirements and fast inference speed during the search process. Table 5 shows that GZAS significantly outperforms GDAS, showing the advantage of our ZARTS framework against the DARTS framework.

**P-ZARTS:** We prune 3/2/1 redundant operations for each super-edge at 20/30/40 epoch during the search process. Also, our P-ZARTS follows the handcrafted criteria of P-DARTS, i.e., fixing the number of skip-connection operations as 2 for normal cells. Table 5 shows that P-ZARTS outperforms P-DARTS by 0.15% accuracy on CIFAR-10 and even slightly surpasses ZARTS, which results from the handcrafted constraint on the number of skip-connection operations in P-DART. Experimental results in the prior work [5] show that the performance of P-DARTS decreases to 96.48% on CIFAR-10 if the handcrafted constraint is removed. The performance of P-ZARTS decreases to 97.20% if the

Table 3: Comparison with prior works that directly search on ImageNet.

| Methods | Params (M) ↓ | Top-1 Err. (%) ↓ | Cost (GPU-days) ↓ |
|---------|--------------|------------------|-------------------|
| SPOS [8] | 3.5 | 25.6 | 12 |
| ProxylessNAS [2] | 7.1 | 24.9 | 8.3 |
| FBNet-C [15] | 5.5 | 25.1 | 9 |
| **ZARTS** | 5.2 | 24.4 | 2.6 |
| **MergeZARTS** | 5.5 | 24.3 | 0.7 |

handcrafted constraint is removed, which still outperforms P-DARTS, demonstrating the effectiveness of our ZARTS framework.

**MergeZARTS:** We adopt the weight merge technique in MergeNAS by sharing weights among the convolutions on one super-edge and merging them into one. Such a strategy can reduce the GPU memory requirements and save the computation resource. Unlike P-ZARTS, which adopts a handcrafted constraint on the number of skip-connection, MergeNAS and MergeZARTS have no specific constraints. Table 5 shows that MergeZARTS significantly outperforms MergeNAS, demonstrating the effectiveness of our ZARTS framework. Moreover, we observe MergeZARTS also surpasses ZARTS by nearly 0.3% accuracy on CIFAR-10, which can result from the fewer redundant network weights $\omega$ in the supernet. [13] analyze that the supernet is an over-parameterized network, whose weights $\omega$ is hard to converge after only training for 50 epochs. The weight merge technique can reduce the redundant weights, alleviating the difficulty to train weights in the supernet, which can also benefit our ZARTS framework: Faster convergence ability make it easier and more accurate to approximate the optimal network weights $\omega^*(\alpha)$ by gradient descent ($M$ iterations in our experiments).

## 2.7 Directly Search on ImageNet

ZARTS has the same memory cost as DARTS and it can be reduced by combining ZARTS with other orthogonal variants, such as MergeNAS (please see Table 5 in the submission). ZARTS can also directly search on ImageNet on a single NVIDIA 3090 GPU with 24G memory. Specifically, we train a supernet with 8 cells and 16 initial channels for 50 epochs with batch size 128. For MergeZARTS, the memory-efficient variants of ZARTS introduced in Sec. 5.3, we can train the supernet with batch size 256. To reduce search time, we randomly sample 25% samples from the training set of ImageNet and divided it into two subsets to train weights and architecture parameters respectively. Performance of the discovered architectures and the search cost is shown in Table 3.

## 2.8 Experiments on NAS-Bench-201

we also search on NAS-Bench-201 and report the results in Table 4. We adopt the hyperparameters in NAS-Bench-201 for fair comparison. The results are averaged over three independent runs. Our method outperforms ENAS, DARTS, and ENAS on three datasets. The accuracy curve is also plotted in Fig. 4, showing that the search process of ZARTS is pretty stable.

## 2.9 Visualization of Architectures

Note that in all our experiments, we directly utilize the architecture at the final epoch (epoch 50) as the inferred network. No model selection procedure is needed.

We visualize the architectures of normal and reduction cells searched by ZARTS in DARTS's search space on CIFAR-10, as is shown in Fig. 5. The architecture searched on CIFAR-100 dataset is illustrated in Fig. 6. We also conduct experiments in the four difficult search spaces of RDARTS [17] on CIFAR-10 [9], CIFAR-100 [9], and SVHN [11]. The searched architectures are illustrated in Fig. 7, Fig. 8, and Fig. 9.

Moreover, we plot the architectures of normal and reduction cells derived every 25 epochs in Fig. 10 and Fig. 11. The discovered architectures vary during the first 100 epochs, and become stable after that, and the topology remains unchanged after 150 epochs.

Table 4: Top-1 test accuracy (%) for classification on NAS-Bench-201. The results of other architectures are obtained from the paper of NAS-Bench-201. The performance of our method is averaged over three independent runs.

| Methods | CIFAR-10 | | CIFAR-100 | | ImageNet-16-120 | |
|---|---|---|---|---|---|---|
| | valid | test | valid | test | valid | test |
| RSPS | 80.42±3.58 | 84.07±3.61 | 52.12±5.55 | 52.31±5.77 | 27.22±3.24 | 26.28±3.09 |
| DARTS-V1 | 39.77±0.00 | 54.30±0.00 | 15.03±0.00 | 15.61±0.00 | 16.43±0.00 | 16.32±0.00 |
| DARTS-V2 | 39.77±0.00 | 54.30±0.00 | 15.03±0.00 | 15.61±0.00 | 16.43±0.00 | 16.32±0.00 |
| GDAS | 89.89±0.08 | 93.61±0.09 | 71.34±0.04 | 70.70±0.30 | 41.59±1.33 | 41.71±0.98 |
| SETN | 84.04±0.28 | 87.64±0.00 | 58.86±0.06 | 59.05±0.24 | 33.06±0.02 | 32.52±0.21 |
| ENAS | 37.51±3.19 | 53.89±0.58 | 13.37±2.35 | 13.96±2.33 | 15.06±1.95 | 14.84±2.10 |
| **ZARTS** | 91.23±0.24 | 93.98±0.27 | 71.64±1.31 | 71.67±1.30 | 44.46±1.36 | 45.06±0.97 |
| ResNet | 90.83 | 93.97 | 70.42 | 70.86 | 44.53 | 43.63 |
| Optimal | 91.61 | 94.37 | 73.49 | 73.51 | 46.77 | 47.31 |

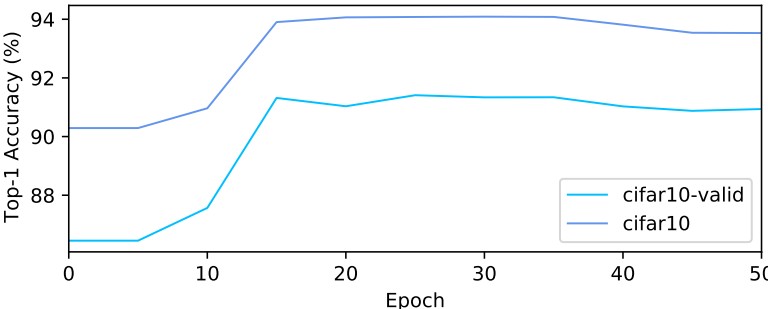

Figure 4: Accuracy curve of the searched architecture during the search process on NAS-Bench-201.

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

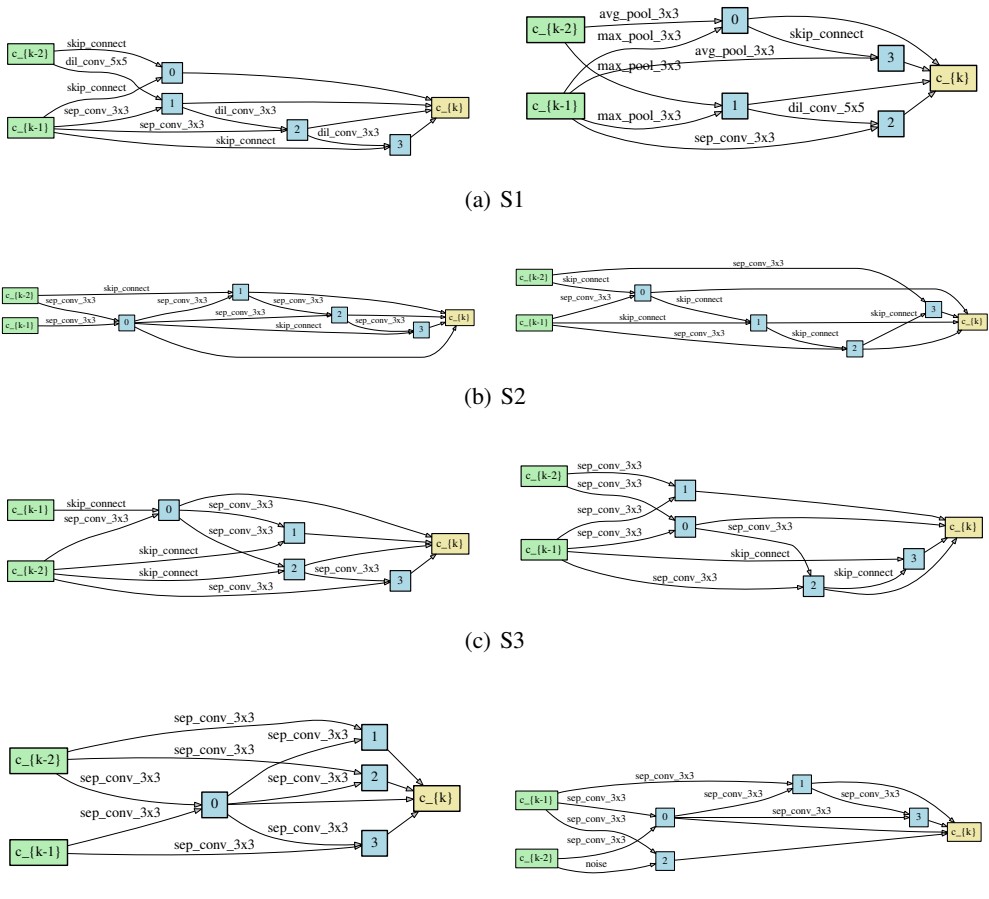

(a) S1

(b) S2

(c) S3

(d) S4

Figure 7: The architectures of normal and reduction cells searched by ZARTS on CIFAR-10 in the four difficult search space of RDARTS. The left column shows the normal cells, while the right column shows the reduction cells.

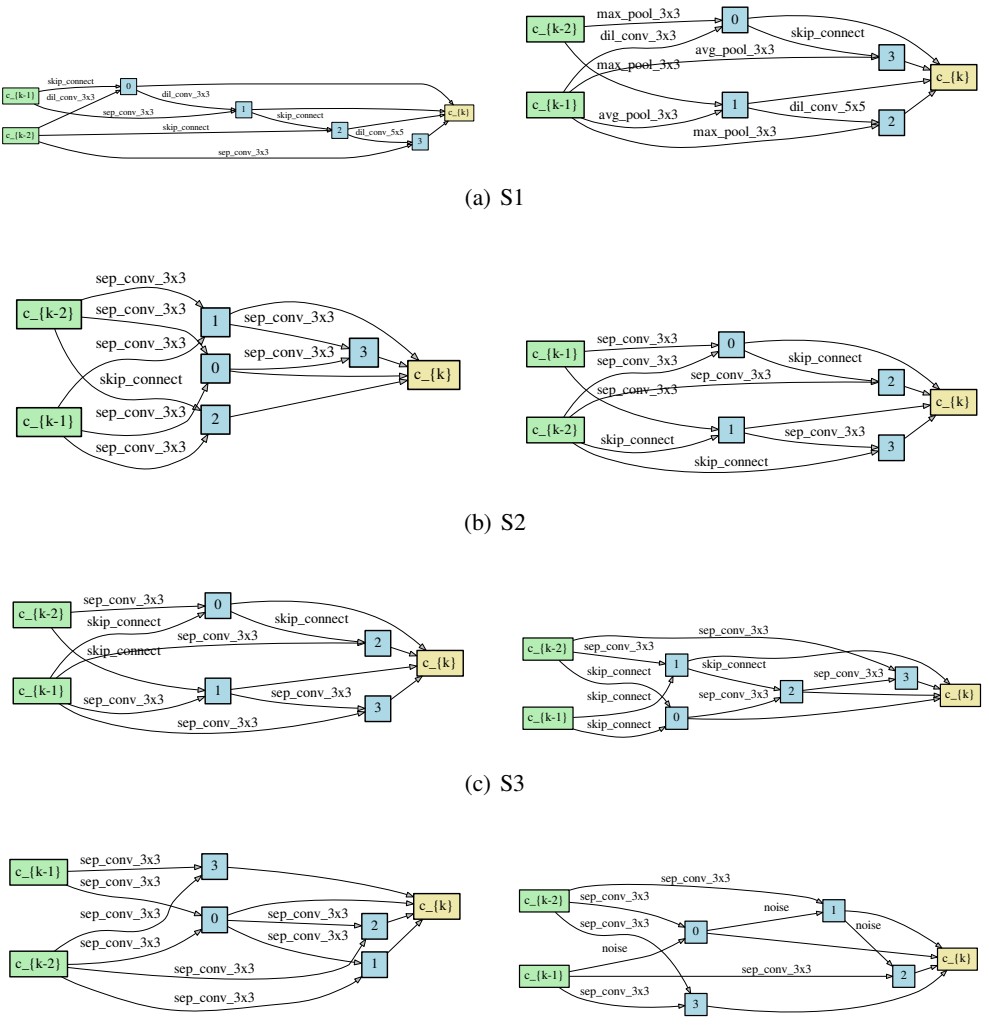

(a) S1

(b) S2

(c) S3

(d) S4

Figure 8: The architectures of normal and reduction cells searched by ZARTS on CIFAR-100 in the four difficult search space of RDARTS. The left column shows the normal cells, while the right column shows the reduction cells.

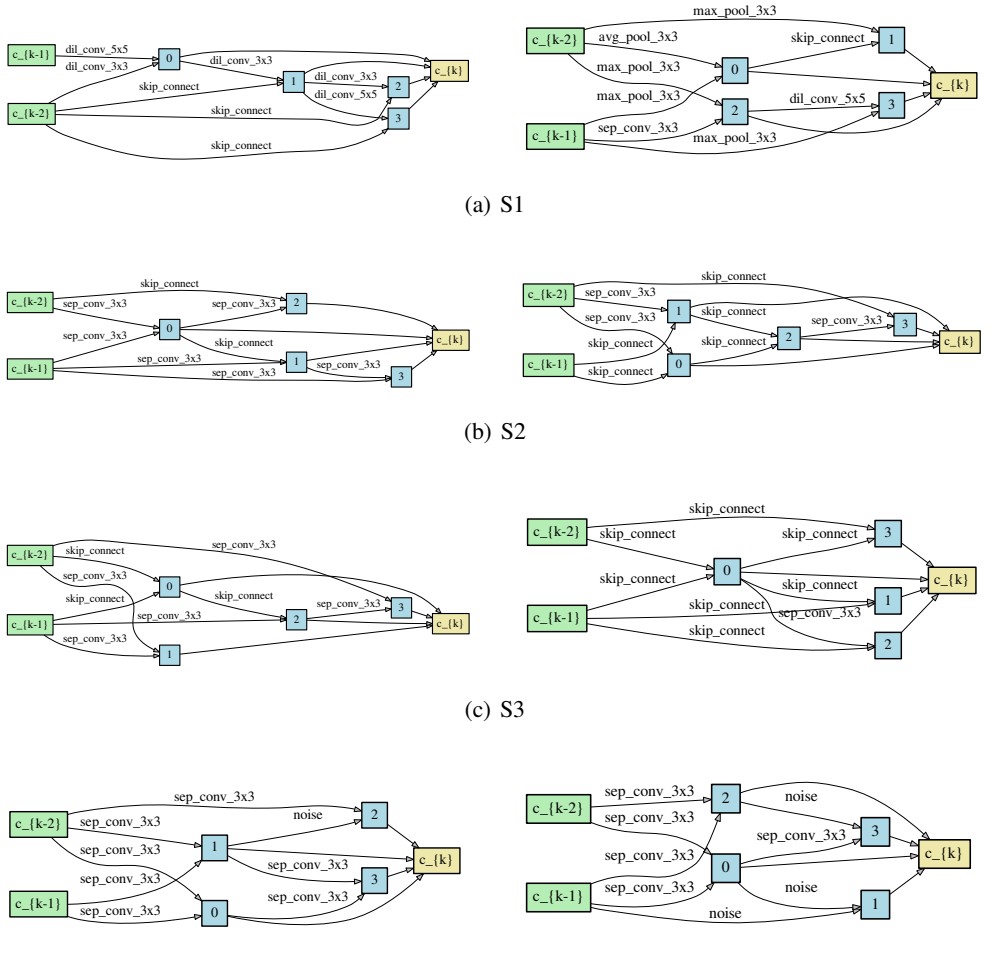

(a) S1

(b) S2

(c) S3

(d) S4

Figure 9: The architectures of normal and reduction cells searched by ZARTS on SVHN in the four difficult search space of RDARTS. The left column shows the normal cells, while the right column shows the reduction cells.

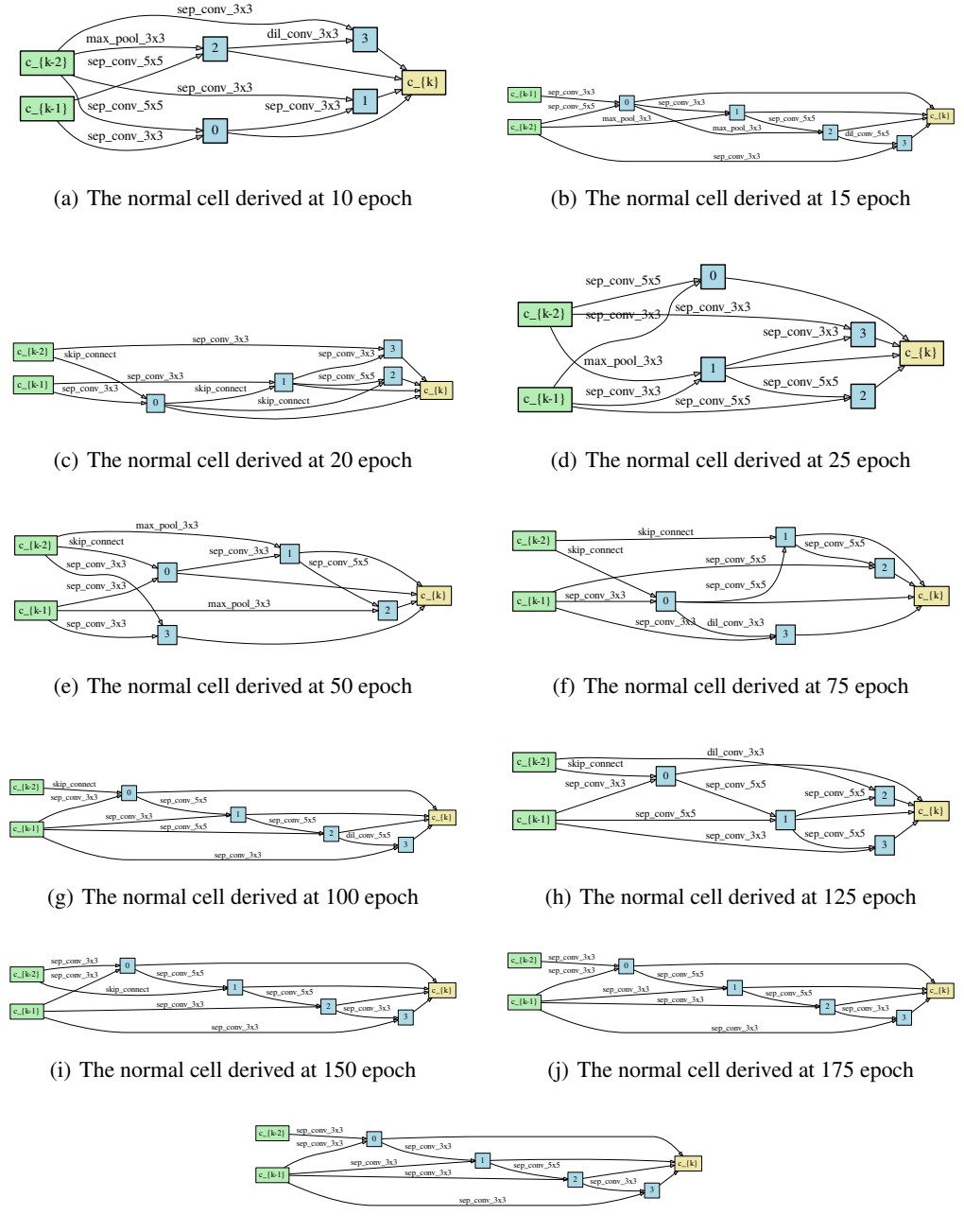

(a) The normal cell derived at 10 epoch

(b) The normal cell derived at 15 epoch

(c) The normal cell derived at 20 epoch

(d) The normal cell derived at 25 epoch

(e) The normal cell derived at 50 epoch

(f) The normal cell derived at 75 epoch

(g) The normal cell derived at 100 epoch

(h) The normal cell derived at 125 epoch

(i) The normal cell derived at 150 epoch

(j) The normal cell derived at 175 epoch

(k) The normal cell derived at 200 epoch

Figure 10: The derived architectures of normal cell every 25 epochs, which are searched by ZARTS on CIFAR-10 for 200 epochs.

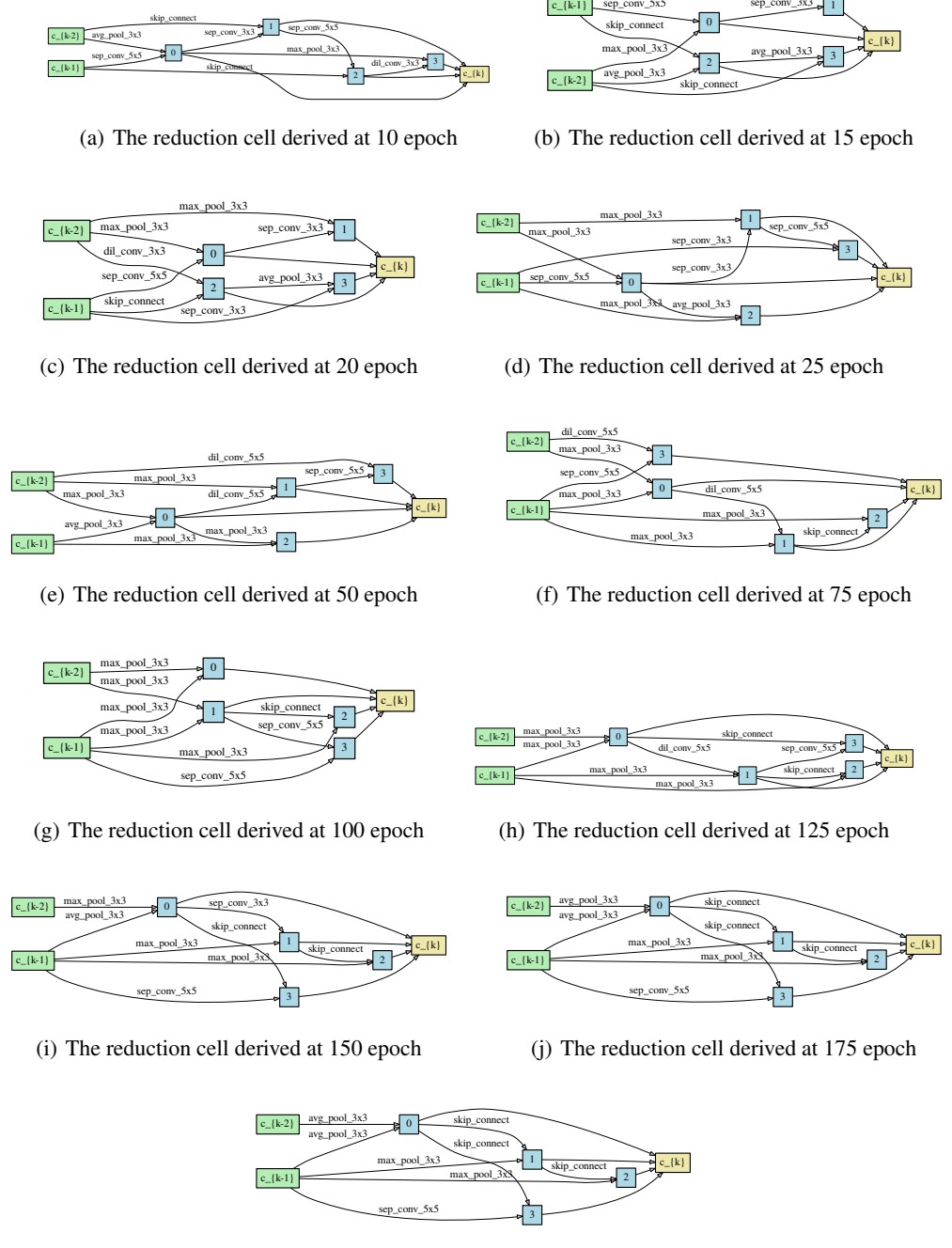

(a) The reduction cell derived at 10 epoch

(b) The reduction cell derived at 15 epoch

(c) The reduction cell derived at 20 epoch

(d) The reduction cell derived at 25 epoch

(e) The reduction cell derived at 50 epoch

(f) The reduction cell derived at 75 epoch

(g) The reduction cell derived at 100 epoch

(h) The reduction cell derived at 125 epoch

(i) The reduction cell derived at 150 epoch

(j) The reduction cell derived at 175 epoch

(k) The reduction cell derived at 200 epoch

Figure 11: The derived architectures of reduction cell every 25 epochs, which are searched by ZARTS on CIFAR-10 for 200 epochs.