# OpenReview forum: "ZARTS: On Zero-order Optimization for Neural Architecture Search"
_NeurIPS.cc/2022/Conference — NeurIPS 2022 Accept_

### Official Review · Reviewer_PVAt · 2022-07-10

**Rating:** 8
**Confidence:** 5
**Soundness:** 3 good
**Presentation:** 3 good
**Contribution:** 4 excellent

**Summary:**

This paper proposes a new neural architecture search method, namely ZARTS, which solves the bi-level optimization through zero-order optimization. It provides interesting analysis on the differentiable assumption in DARTS and show that the assumption can distort the loss landscape and misleads the search target in Fig. 1. Then this work discards the differentiable assumption and turns to zero-order optimization techniques which to my knowledge have not been used in neural architecture search (NAS) literature. This work also shows the connection between ZARTS and DARTS and points out that ZARTS degrades to the first/second-order DARTS when the iteration number M=0 (for first-order DARTS) or M=1 (for second-order DARTS). The well-designed experiments on the search spaces of DARTS and RobustDARTS verify the effectiveness of ZARTS which are implemented in three variants with the tradeoff between efficiency and efficacy. The paper is technically clear and well written.

**Questions:**

1. This work introduces to apply three zero-order optimization methods to NAS. I wonder what properties for zero-order optimization methods are necessary to apply to NAS task? This question is also pertinent to the novelty of the paper, for better motivating the adoption of three specific zero-order optimization techniques presented in the paper.
2. Zero-order optimization methods are known more inefficient than gradient descent method. It would be better if the authors could provide more discussion on why ZARTS converges better than DARTS?


**Limitations:**

I think there is no particular limitation of this work.

**Strengths And Weaknesses:**

Strengths:

++ This paper provides new and interesting analysis on the limitations of differentiable assumption in DARTS, showing that it can distort the loss landscape resulting in the severe instability issue (in Fig. 1a).

++ By showing the drawbacks of differentiable assumption, it is a good idea to adopt zero-order optimization methods to solve the bi-level optimization problem in NAS.

++ The theoretical analysis on the connection between ZARTS and DARTS are worth publishing, specifically: when the iteration number M is set as 0/1, ZARTS degrades to the first/second-order DARTS.

++ Experiments verify the efficacy of ZARTS, showing that zero-order optimization can effectively alleviate the instability issue of DARTS.

++ This paper provides three variants of ZARTS by combining with three variants of DARTS. Experiments indicate that the variants can speed up the search process and further improve the performance, showing the potential of zero-order optimization methods for NAS.

Weaknesses:

-- Zero-order optimization methods are known more inefficient than gradient descent method, so that ZARTS requires 1 GPU-day to search, which is more time-consuming than other DARTS-based methods.

---

> ### Author Response · Authors · 2022-08-01
> **Response to Reviewer PVAt**
>
> Thank you for your thorough and valuable comments. We answer your questions as follows in the hope of resolving your concerns.
>
> **Q1: ZARTS is more time-consuming than other DARTS-based methods.**
>
> A1: We agree that the search cost of ZARTS is higher than some recent works aiming to speed up the search process, but it is more efficient than other methods that aim to stabilize DARTS, such as RDARTS, S-DARTS, and Amended-DARTS (as compared in Table 3). This work focuses on revealing the neglected harm of inaccurate approximation for optimal network weights $\boldsymbol{\omega^*(\boldsymbol{\alpha})}$ in DARTS and proposes to address the bi-level optimization problem in NAS by zero-order optimization methods.
>
> Moreover, our method is orthogonal to the prior works aiming to speed up the search process, such as P-DARTS, GDAS and MergeNAS. Sec. 5.3 and Table 5 in the submission show that ZARTS can be sped up by combining with these variants.
>
>
> **Q2: What properties for zero-order optimization methods are necessary to apply to NAS task?**
>
> A2: NAS requires zero-order optimization methods with efficient sampling strategies. We apply three zero-order optimization methods, including Multi-point estimator (RS), maximum-likelihood guided parameter search (MGS), and gradientLess descent (GLD), with progressive computational complexity and performance. Table 2 compares the three presented zero-order solvers (RS, MGS, and GLD), showing that sampling strategy significantly affects the performance. RS randomly samples candidates $\boldsymbol{u}_i$ and thus has the least computational complexity, while GLD has to sample on spheres with various radii (line 173) and thus has the most computational complexity but achieves the best in Table 2. MGS, however, can trade accuracy off against speed. We hope the empirical results and analysis in this work could shed light on the frontier of NAS, and we will leave the exploration of new zero-order optimization methods as our future work.
>
>
> **Q3: Why does ZARTS converge better than DARTS?**
>
> A3: The main reason lies in that ZARTS circumvents the 1st-order approximation of $\boldsymbol{\omega}^*(\boldsymbol\alpha)$ and can search on the actual loss landscape. In contrast, DARTS adopts the 1st-order approximation, which distorts the loss landscape and optimum (please see Fig. 1 in the submission). Fig. 1c further verifies our analysis. We apply DARTS and ZARTS by starting from the same initial point and updating $\boldsymbol{\alpha}$ 10 times. ZARTS converges to the optimum, but DARTS does not.

---

### Official Review · Reviewer_eA5X · 2022-07-11

**Rating:** 7
**Confidence:** 4
**Soundness:** 3 good
**Presentation:** 4 excellent
**Contribution:** 3 good

**Summary:**

The authors reexamine the bi-level optimization of NAS and reveal that the differentiable assumption in DARTS can mislead the search process. To verify the analysis, they plot loss landscapes of DARTS and show that first-order approximation of network weight will distort the landscape and drift the optimal. To tackle such an issue, the authors propose ZARTS by utilizing zero-order optimization for architecture parameters. Specifically, they adopt three different zero-order optimizers and sufficient empirical evaluation demonstrates that zero-order optimization unanimously outperforms DARTS. Besides, the authors provide extensive convergence experiments to show the stability of ZARTS. In general, ZARTS provides a new insight for NAS task and achieves good performance on multiple search spaces and datasets.

**Questions:**

Please see the weakness. Besides, I notice that the variants of ZARTS, i.e., mergeZARTS and GZAS, requires fewer GPU memory, so my question is, can ZARTS directly search on ImageNet?

**Limitations:**

The limitations were discussed.

**Strengths And Weaknesses:**

Strengths:
1. The paper is well-written and the motivation is clear. The authors first reexamine the bi-level optimization of NAS, then analyze the fundamental limitations in DARTS, and lead to the introduction of ZARTS.
2. Illustration in Fig.1 is convincing, which provides enough evidence on the drawback of differentiable assumption of DARTS and the advantage of ZARTS against DARTS.
3. Apart from applying three zero-order optimization methods to NAS, the authors also theoretically show that ZARTS can degrade to DARTS under differentiable assumption.
4. The experimental results are impressive, showing that ZARTS outperforms DARTS and its variants on multiple search spaces and datasets. Besides, the authors conduct sufficient evaluation about the stability of ZARTS.

Weaknesses:
1. The authors base their zero-order search methods on three existing zero-order optimization algorithms. Of course it is fine with the current excellent analysis and nontrivial technical adaptation to the NAS problem, while it will be more impressive if the authors could propose their own zero-order algorithm tailored to NAS. I know proposing a new zero-order optimizer it self could be a prominent paper, so at least the authors could provide more discussion on this point.
2. The searching cost of ZARTS seems to be longer than other variants of DARTS (in Table 3) Can the authors provide more detailed discussion on the searching cost?

Overall, the motivation is clear and the evaluation is sufficient. The major concern is the search cost of ZARS, and I hope the authors could provide more analysis in the rebuttal.

---

> ### Author Response · Authors · 2022-08-01
> **Response to Reviewer eA5X**
>
> Thank you for your time and constructive feedback, we answer your questions as follows, which we hope will resolve your concerns.
>
> **Q1: Discussion on new zero-order optimization methods for NAS.**
>
> A1: Thanks for your suggestion. Inventing a new zero-order solver is intellectually attractive, yet our work is more focused on understanding the reason (so far unclear) why zero-order optimization outperforms first-order gradient descent for the (non-differentiable) bi-level optimization problem in NAS task, which mainly lies in an accurate estimation for $\boldsymbol{\omega}^*(\boldsymbol{\alpha})$ (please see discussion in Sec. 3 and Fig. 1). Results in Table 2 indicate that even vanilla multi-point estimator (RS) surpasses DARTS, empirically verifying our analysis on the superiority of zero-order optimization against gradient-based methods in NAS task.
>
> By comparing the three presented zero-order solvers (RS, MGS and GLD), we observe that sampling strategy significantly affects the performance. RS randomly sample candidates $\boldsymbol{u}_i$ and thus has the most minor computational complexity. In contrast, GLD has to sample on spheres with various radii (line 173) and thus has the most computational complexity but achieves the best in Table 2. MGS, however, can trade off the accuracy and speed. We hope the experimental results and analysis in this work could shed light on the frontier of NAS. We will leave the exploration of new zero-order optimization methods as our future work.
>
> **Q2: Discussion on the searching cost of ZARTS.**
>
> A2: ZARTS and DARTS-2nd have similar search cost (1.0 GPU-day) but ZARTS performs much better due to accurate estimation for $\boldsymbol{\omega}^*(\boldsymbol{\alpha})$. Though ZARTS is twice slower than DARTS-1st, it can be sped up by combining with other orthogonal methods, such as GDAS, P-DARTS and MergeNAS. Sec. 5.3 and Table 5 in the submission show the results of these ZARTS variants. Specifically, GZAS (ZARTS+GDAS) achieves 97.34\% average performance with only 0.3 GPU-day; P-ZARTS (ZARTS+P-DARTS) achieves 97.59\% average performance with 0.4 GPU-day; MergeZARTS (ZARTS+MergeNAS) achieves 97.64\% average performance with 0.5 GPU-day.
>
> **Q3: Can ZARTS directly search on ImageNet?**
>
> A3: Yes. ZARTS has the same memory cost as DARTS, and it can be reduced by combining ZARTS with other orthogonal variants, such as MergeNAS (please see Table 5 in the submission). ZARTS can also directly search on ImageNet on a single NVIDIA 3090 GPU with 24G memory. Specifically, we train a supernet with 8 cells and 16 initial channels for 50 epochs with batch size 128. For MergeZARTS, the memory-efficient variants of ZARTS introduced in Sec. 5.3, we can train the supernet with batch size 256. To reduce search time, we randomly sample 25\% samples from the training set of ImageNet and divide it into two subsets to train weights and architecture parameters, respectively. The performance of the discovered architectures and the search cost is shown in Table 4 in the supplementary material. We also briefly list our results here.
>
> | Method       | Params (M) | Top-1 Error (%) | Search Cost (GPU-day) |
> |:------------ | ----------:|:---------------:| --------------------- |
> | SPOS         |        3.5 |      25.6       | 12                    |
> | ProxylessNAS |        7.1 |      24.9       | 8.3                   |
> | FBNet-C      |        5.5 |      25.1       | 9                     |
> | ZARTS        |        5.2 |      24.4       | 2.6                   |
> | MergeZARTS   |        5.5 |      24.3       | 0.7                   |

---

### Official Review · Reviewer_LjFh · 2022-07-12

**Rating:** 5
**Confidence:** 3
**Soundness:** 3 good
**Presentation:** 3 good
**Contribution:** 2 fair

**Summary:**

This paper proposes a DARTS variant named ZARTS, which replace the first-order and second-order approximations with zero-order approximations. Experiments on various datasets are conducted to show the superiority of ZARTS.

**Questions:**

1. Does ZARTS has the same memory cost as DARTS? Can ZARTS directly search on ImageNet?

2. Will the zero-order optimization still be accurate when searching on a huge search space?

**Limitations:**

Discussion of limitations was provided.

**Strengths And Weaknesses:**

Strengths:
1. The inaccurate gradient estimation in first-order approximation could result in severe performance collapse and instability problems, this should be highlight in the community. This paper proposes a zero-order approximation method, which seems has better architecture optimization according to the paper's visualizations.
2. According to Table 5, ZARTS applies to various DARTS variants, and can bring consistent improvements to them.

Weaknesses:
1. The search cost of ZARTS is similar to DARTS (2nd). Many variants adopt first-order approximation instead of the second-order approximation for faster search, though it has inaccurate gradient estimations; while the proposed zero-order approximation has a similar cost as the second-order one, which may be a major limitation of this work.
2. Figure 1 is not sufficient enough to show the superiority of ZARTS in architecture optimization. I suggest the authors to conduct experiments on NAS benchmarks (e.g., NAS-Bench-201) to show the accuracy curves of architectures during the searching procedure.

---

> ### Author Response · Authors · 2022-08-01
> **Response to Reviewer LjFh [1/2]**
>
> Thank you for your valuable comments. We answer the questions as follows in the hope of resolving your concern.
>
> **Q1: The search cost of ZARTS is similar to DARTS (2nd), which may be a major limitation of this work.**
>
> A1: Though ZARTS has a similar search cost as DARTS (2nd), ZARTS considers $M=10$ steps gradient descent to estimate optimal network weights $\boldsymbol{\omega}^*(\boldsymbol{\alpha})$, while DARTS (2nd) only considers one step gradient descent for $\boldsymbol{\omega}$. Therefore, ZARTS is more efficient than DARTS.
> Moreover, we can simply speed up ZARTS by reducing $M$. In particular, when $M=2$, ZARTS achieves 97.38\% accuracy, still outperforming DARTS (2nd) with less search cost (0.3 GPU-day), as shown in Table 1 (right) in the supplementary material. We also briefly list our results here.
> | Model      | ZARTS(M=2) | ZARTS(M=5) | ZARTS(M=8) | ZARTS(M=10) | DARTS(1st) | DARTS(2nd) |
> |:---------- | ---------:|:---------:| --------- | ---------- | --------- | --------- |
> | Error (%)  |    2.62 |  2.60   |    2.57       |        2.54    |     3.00      |    2.76       |
> | Cost (GPU-day)           |    0.3       |   0.6        |    0.8       |    1.0        |  0.4         |   1.0        |
>
>
> **Q2: Many variants adopt first-order approximation instead of the second-order approximation for faster search, though it has inaccurate gradient estimations.**
>
> A2: The performance of these variants can be further improved by combining with ZARTS to refine the inaccurate gradient estimations, as verified by Table 5 in the submission. Specifically, in Sec. 5.3, we combine ZARTS with three variants of DARTS and derive GZAS, P-ZARTS and MergeZARTS, which achieve better performance than the original methods in only 0.5 GPU-day, showing that their fast search speed can also be maintained.
>
> **Q3: Suggestion about searching on NAS-Bench-201 to show the accuracy curves of architectures during the searching procedure.**
>
> A3: Thanks for your suggestion. On the one hand, Fig. 2 in the submission shows the accuracy curves of architectures searched on DARTS's search space. Specifically, we first train the supernet for 200 epochs by DARTS and ZARTS and obtain the discovered architectures every 25 epochs. Then we train those discovered architectures from scratch for 600 epochs in the same experimental settings. Please see Line 254-267 in the submission and Sec.2.5 in the supplementary material for details of experimental settings.). We believe that Fig. 2 demonstrates the superiority of ZARTS in architecture optimization. We observe that the architectures searched by ZARTS perform stably well (around 97.40\% accuracy), while the performance
> of those searched by DARTS gradually drops. Moreover, the parameter number of architectures searched by DARTS decreases significantly after 50 epochs, indicating that parameterless operations dominate the topology and the instability issue occurs.
>
> On the other hand, we search on NAS-Bench-201 and report the results in Table 4 in the supplementary material. We also briefly list our results here. Specifically, we adopt the hyperparameters in NAS-Bench-201 for a fair comparison. The results are averaged over three independent runs. Our method outperforms ENAS, DARTS, and ENAS on three datasets. The accuracy curve is also plotted in Fig. 4 in the supplementary material, showing that the search process of ZARTS is pretty stable.
> | Method     | CIFAR-10(valid) | CIFAR-10(test) | CIFAR-100(valid) | CIFAR-100(test) | ImageNet16-120(valid) | ImageNet16-120(test) |
> |:---------- | ---------------:|:--------------:| ---------------- | --------------- | --------------------- | -------------------- |
> | DARTS(1st) |  $39.77\pm0.00$ | $54.30\pm0.00$ | $15.03\pm0.00$   | $15.61\pm0.00$  | $16.43\pm0.00$        | $16.32\pm0.00$       |
> | DARTS(2nd) |  $39.77\pm0.00$ | $54.30\pm0.00$ | $15.03\pm0.00$   | $15.61\pm0.00$  | $16.43\pm0.00$        | $16.32\pm0.00$       |
> | GDAS       |  $89.89\pm0.08$ | $93.61\pm0.09$ | $71.34\pm0.04$   | $70.70\pm0.30$  | $41.59\pm1.33$        | $41.71\pm0.98$       |
> | **ZARTS**           |   $91.23\pm0.24$              |   $93.98\pm0.27$             | $71.64\pm1.31$                 |   $71.67\pm1.30$              |     $44.46\pm1.36$                  |  $45.06\pm0.97$                    |

---

> > ### Author Response · Authors · 2022-08-01
> > **Response to Reviewer LjFh [2/2]**
> >
> > **Q4: Does ZARTS has the same memory cost as DARTS? Can ZARTS directly search on ImageNet?**
> >
> > A4: ZARTS has the same memory cost as DARTS, and it can be reduced by combining ZARTS with other orthogonal variants, such as MergeNAS (please see Table 5 in the submission). ZARTS can also directly search on ImageNet on a single NVIDIA 3090 GPU with 24G memory. Specifically, we train a supernet with 8 cells and 16 initial channels for 50 epochs with batch size 128. For MergeZARTS, the memory-efficient variants of ZARTS introduced in Sec. 5.3, we can train the supernet with batch size 256. To reduce search time, we randomly sample 25\% samples from the training set of ImageNet and divide it into two subsets to train weights and architecture parameters, respectively. The performance of the discovered architectures and the search cost is shown in Table 3 in the supplementary material. We also briefly list our results here.
> >
> > | Method       | Params (M) | Top-1 Error (%) | Search Cost (GPU-day) |
> > |:------------ | ----------:|:---------------:| --------------------- |
> > | SPOS         |        3.5 |      25.6       | 12                    |
> > | ProxylessNAS |        7.1 |      24.9       | 8.3                   |
> > | FBNet-C      |        5.5 |      25.1       | 9                     |
> > | **ZARTS**        |        5.2 |      24.4       | 2.6                   |
> > | **MergeZARTS**   |        5.5 |      24.3       | 0.7                   |
> >
> >
> > **Q5: Will the zero-order optimization still be accurate when searching on a huge search space?**
> >
> > A5: ZARTS is a general search method like DARTS and can be transferred to other search spaces, and our estimation for $\boldsymbol{\omega}^*(\boldsymbol{\alpha})$ will still be more accurate than DARTS. The primary concern about searching on a huge search space lies in the vast GPU memory requirement to build a supernet, which is a pervasive issue for all one-shot based NAS methods. Fortunately, our ZARTS can be easily combined with other memory-efficient methods, such as GDAS and MergeNAS, which can reduce more than half of the GPU memory requirement (please see Table 5 in the submission).

---

> > > ### Comment · Reviewer_LjFh · 2022-08-03
> > > **Response to Authors**
> > >
> > > Thanks for your detailed response, most of my concerns are addressed. I have raised my rating to borderline accept.

---

### Meta-Review · Area_Chair_FNko · 2022-08-26

**Recommendation:** Accept
**Confidence:** Certain

**Metareview:**

This paper aims to solve the instability issues of differentiable architecture search (DARTS) using zero-order optimization. Three different optimization techniques are proposed and their efficacy is demonstrated successfully on several benchmark datasets and different variants of DARTS. Although there are some concerns regarding the computational complexity of zero-order optimization, the reviewers have found the contribution of this submission significant for acceptance at NeurIPS. Given this, we are happy to recommend acceptance.

**Award:**

No

---

### Decision · Program_Chairs · 2022-09-14

Accept